# Reduced *FOXF1* links unrepaired DNA damage to pulmonary arterial hypertension

Sarasa Isobe [1,2,3,4], Ramesh V. Nair[5], Helen Y. Kang[1,6], Lingli Wang[1,2,3,4], Jan-Renier Moonen[1,2,3,4], Tsutomu Shinohara[1,2,3,4], Aiqin Cao[1,2,3,4], Shalina Taylor[2,3,4], Shoichiro Otsuki [2,3,4], David P. Marciano[3,6], Rebecca L. Harper [2,3,4], Mir S. Adil [1,2,3,4], Chongyang Zhang[1,2,3,4], Mauro Lago-Docampo [1,2,3,4], Jakob Körbelin [7], Jesse M. Engreitz [1,3,6], Michael P. Snyder [3,6] & Marlene Rabinovitch [1,2,3,4] ✉

Pulmonary arterial hypertension (PAH) is a progressive disease in which pulmonary arterial (PA) endothelial cell (EC) dysfunction is associated with unrepaired DNA damage. *BMPR2* is the most common genetic cause of PAH. We report that human PAEC with reduced BMPR2 have persistent DNA damage in room air after hypoxia (reoxygenation), as do mice with EC-specific deletion of *Bmpr2* (EC-*Bmpr2*[-/-]) and persistent pulmonary hypertension. Similar findings are observed in PAEC with loss of the DNA damage sensor *ATM*, and in mice with *Atm* deleted in EC (EC-*Atm*[-/-]). Gene expression analysis of EC-*Atm*[-/-] and EC-*Bmpr2*[-/-] lung EC reveals reduced *Foxf1*, a transcription factor with selectivity for lung EC. Reducing *FOXF1* in control PAEC induces DNA damage and impaired angiogenesis whereas transfection of *FOXF1* in PAH PAEC repairs DNA damage and restores angiogenesis. Lung EC targeted delivery of *Foxf1* to reoxygenated EC-*Bmpr2*[-/-] mice repairs DNA damage, induces angiogenesis and reverses pulmonary hypertension.

Pulmonary arterial hypertension (PAH) is a disease characterized by progressive elevation in pulmonary arterial pressure that culminates in right heart failure due to narrowing and obstruction of pulmonary vessels owing to pulmonary arterial endothelial dysfunction, exuberant proliferation of smooth muscle-like cells and an inflammatory infiltrate. Although currently approved drugs are vasodilators with limited efficacy in reversing the underlying pathology, agents that can effectively modify the vascular changes are now under consideration for patients with this disease[1]. Genetic analysis of PAH families revealed a heterozygous germline mutation in the gene encoding bone morphogenic protein receptor type 2 (*BMPR2*), a member of the transforming growth factor-β (TGFB) superfamily[2]. About 70-80% of familial PAH and 10-20% of idiopathic PAH cases have mutations in *BMPR2*[3], and *BMPR2* expression is decreased in pulmonary artery (PA) endothelial cells (EC) of PAH patients with or without a *BMPR2* mutation[4]. We previously reported that loss of BMPR2 in PAEC causes vulnerability to apoptosis[5], endothelial mesenchymal transition[6], impaired repression of smooth muscle cell (SMC) proliferation[5] and inflammatory cell recruitment[7], which are all features observed in PAH. However, while mutations in *BMPR2* are a risk factor for PAH, incomplete penetrance (only ≈20% of carriers develop the disease)[8] suggests that at least one additional genetic or environmental modifier ('2nd hit') appears necessary to trigger the pathophysiological changes leading to the development of PAH.

[1]Basic Science and Engineering (BASE) Initiative at the Betty Irene Moore Children's Heart Center, Lucile Packard Children's Hospital, Stanford University School of Medicine, Stanford, CA, USA. [2]Vera Moulton Wall Center for Pulmonary Vascular Diseases, Stanford University, Stanford, CA, USA. [3]Stanford Cardiovascular Institute, Stanford University School of Medicine, Stanford, CA, USA. [4]Department of Pediatrics - Cardiology, Stanford University School of Medicine, Stanford, CA, USA. [5]Stanford Center for Genomics and Personalized Medicine, Stanford University School of Medicine, Stanford, CA, USA. [6]Department of Genetics, Stanford University School of Medicine, Stanford, CA, USA. [7]Department of Oncology, Hematology and Bone Marrow Transplantation, University Medical Center Hamburg-Eppendorf, Hamburg, Germany. ✉e-mail: marlener@stanford.edu

Unrepaired DNA damage is a striking feature of both EC and SMC in PAH[9–12]. In SMC, PARP1[10], EYA3[11] and checkpoint kinase-1 (CHK-1)[12] are upregulated to stimulate DNA repair, but this results in proliferation and resistance to apoptosis of SMC with damaged DNA. Moreover, cultured PAEC from PAH patients carry somatic chromosomal deletions that were not found in blood and other lung cell types of the same patient[13]. We previously related unrepaired DNA damage in PAH PAEC to impaired activation of the DNA damage sensing machinery that resulted from aberrant function of peroxisome proliferator-activated receptor gamma (PPARG)[9]. PPARG is activated downstream of BMPR2[5] and BMPR2 also maintains expression of BRCA1[14] and RAD51[15] which are important for genomic stability and appropriate DNA repair. We reported that loss of BMPR2 in PAEC exposed to the oxidant stress of reoxygenation after hypoxia, as occurs in a variety of clinical conditions modeled in animals such as sleep apnea[16] or acute respiratory distress[17], resulted in reduced P53 and mitochondrial DNA damage[18]. P53 plays a critical role in the repair of DNA damage to restore genome stability[19] and our subsequent study showed that P53 forms a complex with PPARG and upregulates genes important for EC regeneration and DNA repair[20]. Restoring p53 with Nutlin-3a prevented the persistence of pulmonary hypertension in mice with *Bmpr2* selectively deleted in EC (SCL-CreER/R26R/*Bmpr2*$^{fl/fl}$)[21] reoxygenated after hypoxia[20]. Taken together, these findings indicate that BMPR2 plays an important role in the DNA repair mechanism as well as in the maintenance of genes necessary for EC homeostasis. However, it has been unclear whether persistent DNA damage in EC and pulmonary arterial hypertension both result from a common mechanism, which could inform new therapeutic avenues.

Therefore, to elucidate the mechanism that links PAH to unrepaired DNA damage, we first established that there was unrepaired DNA damage in EC-*Bmpr2*$^{-/-}$ mice that have persistent pulmonary hypertension after reoxygenation. We then showed that mice lacking the DNA damage sensor *Atm*[22] in EC (EC-*Atm*$^{-/-}$), also have persistent pulmonary hypertension with unrepaired DNA damage after reoxygenation. To explain the association between unrepaired DNA damage and persistent pulmonary hypertension, we compared gene expression in lung EC from EC-*Bmpr2*$^{-/-}$ and EC-*Atm*$^{-/-}$ mice after reoxygenation. Our findings converged on a common mechanism related to a reduction in *Foxf1*, a transcription factor with relative selectivity for lung EC that is normally upregulated in response to DNA damage to induce DNA repair and to stimulate recovery of EC via angiogenesis. These features were confirmed by loss of FOXF1 function in normal human PAEC and the gain of function in PAH PAEC. Delivery of *Foxf1* to the pulmonary vascular EC[23] during reoxygenation of EC-*Bmpr2*$^{-/-}$ mice resulted in the repair of DNA damage, in the restoration of angiogenesis genes required for the regeneration of normal pulmonary arteries, and in the reversal of pulmonary hypertension. Restoration of normal levels of FOXF1 merits consideration as a PAH therapeutic strategy, made attractive both by its selective expression in lung EC and by its important dual mechanisms of action.

## Results

### Loss of BMPR2 and reoxygenation enhance DNA damage in PAEC

To explore the link between loss of BMPR2 and DNA damage, we reduced *BMPR2* in human PAEC by siRNA (si*BMPR2*) vs. control siRNA and subjected the cells to normoxia for 72 h or to reoxygenation (0.5% hypoxia for 48 h followed by room air for 24 h). Loss of BMPR2 in normoxia resulted in DNA damage that was further exaggerated with reoxygenation as judged by an increase in the DNA damage markers γH2AX and phosphorylated RPA (pRPA) when compared to cells transfected with control siRNA (Fig. 1a). More severe DNA damage with loss of *BMPR2* was confirmed by the comet assay, a method for measuring DNA strand breaks[24] (Fig. 1b) and by quantification of nuclear γH2AX and pRPA by confocal immunofluorescence microscopy

(Fig. 1c). These findings might either result from greater susceptibility to DNA damage as previously reported in PAH EC[25] or by inability to repair damaged DNA. Interestingly, exposure to hypoxia for 48 h resulted in a similar level of DNA damage in control and *BMPR2* siRNA transfected cells, but markers of DNA damage reflecting unrepaired DNA were persistently higher in cells with loss of *BMPR2* during reoxygenation periods of 24, 48 and 72 h (Supplementary Fig. 1a–c).

To establish a relationship between DNA damage, loss of BMPR2, and persistent PAH, we used transgenic mice in which *Bmpr2* was deleted selectively in EC (EC-*Bmpr2*$^{-/-}$) (Supplementary Fig. 1d–f), previously shown to have persistent pulmonary hypertension during reoxygenation after hypoxia[18]. The mice were produced by breeding *Cdh5*-CreER, *Rosa*$^{tdTomato}$, and *Bmpr2*$^{fl/fl}$ mice, and conditionally and specifically deleting *Bmpr2* in EC by an 8-day injection of tamoxifen, as described in the Methods. *Cdh5*-CreER/*Rosa*$^{tdTomato}$ littermate mice treated with tamoxifen were used as controls. The resultant EC-*Bmpr2*$^{-/-}$ mice and controls were maintained in normoxia for 7 weeks or were exposed to hypoxia for 3 weeks and returned to room air for 4 weeks (reoxygenation protocol) (Fig. 1d). We found more γH2AX foci in EC of pulmonary arteries in tissue sections of EC-*Bmpr2*$^{-/-}$ mice compared to the control group during normoxia and reoxygenation (Fig. 1e). There was no significant difference in PAEC γH2AX foci when comparing EC-*Bmpr2*$^{-/-}$ and control murine lung tissue after 3 weeks of hypoxia (Supplementary Fig. 1g, h). Interestingly, EC selective *Bmpr2* deletion also affected neighboring SMC, causing accumulation of SMC DNA damage in normoxia and reoxygenation (Fig. 1f). There was no accumulation of DNA damage in EC in the aortas, kidneys, and coronary arteries of EC-*Bmpr2*$^{-/-}$ mice after reoxygenation, and blood tests showed no obvious renal or hepatic dysfunction in EC-*Bmpr2*$^{-/-}$ mice after reoxygenation (Supplementary Fig. 1i, j). These findings suggest that unrepaired DNA damage due to *Bmpr2* deletion in EC is a feature of the lung vasculature.

### Relating DNA damage to persistent PAH

To determine whether there is a cause-and-effect relationship between DNA damage and persistent PAH, we reduced the level of the DNA damage sensor ATM in human PAEC by *ATM* siRNA vs. control siRNA and subjected the cells to normoxia for 72 h or to reoxygenation (0.5% hypoxia for 48 h followed by room air for 24 h). DNA damage was present under normoxia with loss of ATM and was further increased by reoxygenation as judged by western immunoblot for γH2AX and pRPA (Fig. 2a), by length of comet tails (Fig. 2b) and by confocal immunofluorescence of pRPA and γH2AX intensity (Fig. 2c). While there was no difference after exposure to hypoxia, when comparing *ATM* siRNA vs. control siRNA, the accumulation of DNA damage was persistent from 24 h to 72 h (Supplementary Fig. 2a, b), mirroring the effect of reduced BMPR2. It is widely recognized that ATM activates γH2AX[26] to induce DNA damage repair as do other DNA damage response genes, including DNA-PK and ATR[27]. Loss of *ATM* or *BMPR2* in human PAEC increased ATR and DNA-PK in both normoxia and reoxygenation (Supplementary Fig. 2c), indicating that accumulation of DNA damage by loss of *ATM* or *BMPR2* is not fully compensated by the increase in the ATR or DNA-PK DNA repair pathways.

We investigated whether persistent DNA damage seen with reduced *ATM* or *BMPR2* during reoxygenation is associated with oxidative stress, using the DCFDA assay to assess ROS generation. Loss of *ATM* or *BMPR2* significantly albeit modestly increased ROS levels in normoxia and hypoxia when compared to control cells. However, the major increase in ROS levels in all three cell types occurred during the first 15 minutes of reoxygenation and was mostly resolved during the first hour (Supplementary Fig. 2d), with ROS production only slightly increased with reduced *ATM* or *BMPR2* at each time point. This suggests that the persistence of DNA damage is related more to the genotype than to the extent of DNA damage induced by ROS during reoxygenation.

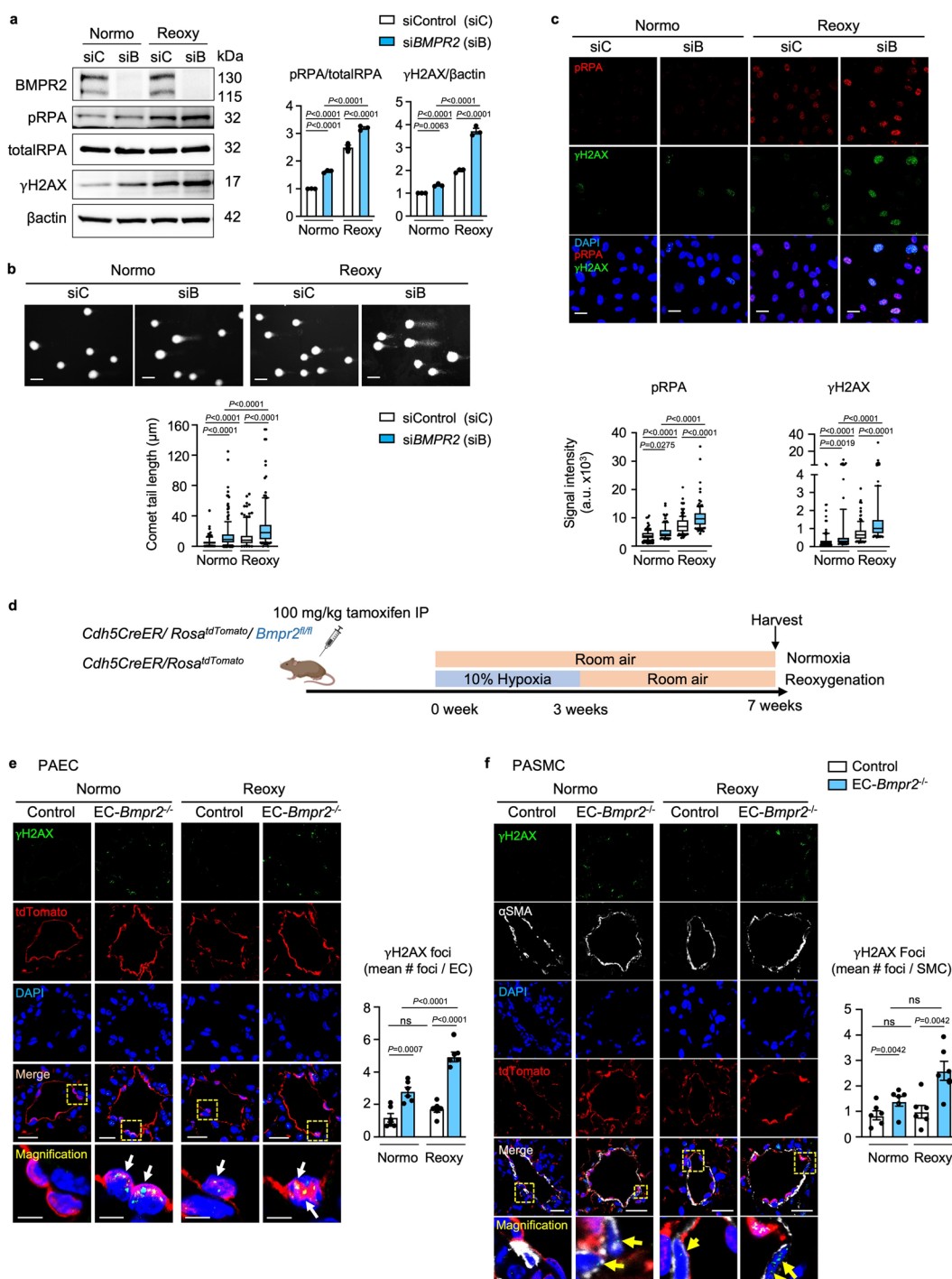

**Fig. 1 | Loss of *BMPR2* results in accumulation of DNA damage in normoxia and after reoxygenation, in cultured cells and in mice with EC-*Bmpr2*⁻/⁻.**
**a–c** Commercially available PAEC transfected with scrambled control siRNA (siC) or siRNA targeting *BMPR2* (siB) were cultured under hypoxia (0.5% O₂) for 48 h followed by room air for 24 h (reoxygenation, reoxy), or room air for 72 h (normoxia, normo). **a** Representative immunoblots of DNA damage markers γH2AX and phosphorylated (p)RPA. *n* = 3 individual experiments. Bars = mean ± S.E.M. *P* values determined by 2-way ANOVA with Holm-Sidak posthoc test. **b** Comet assay shows DNA damage, reflected by comet tail length. Scale bar, 20 μm. *n* = 119, 166, 123, 141 cells for siC normoxia, siB normoxia, siC Reoxy, siB reoxy respectively.
**c** Representative immunohistochemistry of DNA damage markers. Scale bars, 20 μm. *n* = 115, 117, 149, 120 cells for siC normoxia, siB normoxia, siC Reoxy, siB Reoxy, respectively. In (**b**, **c**), box bounds, 25th and 75th percentiles, whiskers 10th to 90th percentiles and box centre shows the median. *P* values were determined by the Kruskal-Wallis ANOVA test with Dunn's test. This experiment was repeated 3

independent times with similar results. **d** *Mouse model*: *Bmpr2* was deleted in EC by injecting tamoxifen to *Cdh5CreER/ Rosa^tdTomato^/Bmpr2^fl/fl^* mice (EC-*Bmpr2*⁻/⁻), bred as in Methods. *Cdh5CreER/Rosa^tdTomato^* mice were used as controls. Mice were exposed to hypoxia (10% oxygen) for 3 weeks followed by 4 weeks of room air (reoxy), or maintained in room air for 7 weeks (normo). Schema created with BioRender.com. **e** γH2AX immunofluorescence foci in PAEC of control and EC-*Bmpr2*⁻/⁻ mice (arrows). Scale bars, 20 μm. Bottom panels: magnified merged image of the area delineated by the dotted line. Scale bars, 5 μm. **f** Mean number of γH2AX immunofluorescent foci in PASMC of control and EC-*Bmpr2*⁻/⁻ mice (arrows). 100–200 μm vessels were examined since control mice had little distal arterial muscularization. αSMA indicates SMC. Scale bars, 20 μm. The bottom panels show magnified merged image of the area delineated by the dotted line. Scale bars, 5 μm. In (**e**, **f**): Data shown as mean ± S.E.M., *n* = 6 mice per group, with 5 vessels analyzed per mouse. *P* values determined by 2-way ANOVA with Holm-Sidak posthoc test. ns, not significant. Source data are provided as a Source Data file.

Next, we used mice with *Atm* deleted in EC (EC-*Atm*⁻/⁻) and exposed them to the normoxia or to the reoxygenation protocol and compared them to control mice under the same conditions (Fig. 2d, Supplementary Fig. 2e, f). EC-*Atm*⁻/⁻ mice were produced by breeding *Cdh5*-CreER, *Rosa*^tdTomato and *Atm*^fl/fl mice, and conditionally and specifically deleting *Atm* in EC by an 8-day injection of tamoxifen, as described in the Methods. *Cdh5*-CreER/*Rosa*^tdTomato littermate mice treated with tamoxifen were used as controls. Unrepaired DNA damage was evident in EC-*Atm*⁻/⁻ vs. control mice during both normoxia and reoxygenation as judged by γH2AX foci (Fig. 2e) whereas there was no difference after hypoxia (Supplementary Fig. 3g, h), in keeping with our observations in the EC-*Bmpr2*⁻/⁻ mice.

Next, we assessed features of pulmonary hypertension in male and female EC-*Atm*⁻/⁻ mice under normoxia, hypoxia, and reoxygenation (Fig. 3a, b, Supplementary Fig. 3a, and Supplementary Table 1). Male and female mice were similar in both control and EC-*Atm*⁻/⁻ mice under normoxia and showed comparable pulmonary hypertension following hypoxia as judged by right ventricular systolic pressure, right ventricular hypertrophy (Fig. 3a, b) and pulmonary artery acceleration time/ejection time (PAAT/ET) (Supplementary Fig. 3c, d). There was a slight reduction in the number of alveolar duct and wall arteries per 100 alveoli without an apparent increase in muscularization of these distal vessels. However, both male and female EC-*Atm*⁻/⁻ mice showed persistent pulmonary hypertension after reoxygenation judged by right ventricular systolic pressure and right ventricular hypertrophy when compared to control mice where indices were similar to those mice maintained under room air conditions (Fig. 3a, b). The persistence of pulmonary hypertension was related to a reduced number of alveolar duct and wall arteries per 100 alveoli in EC-*Atm*⁻/⁻ vs. control mice and an increase in the percent of muscular arteries at these levels (Fig. 3a, b). We did not observe other pulmonary vascular abnormalities, such as occlusion, plexiform lesions, or arteriovenous malformations (Fig. 3c). To investigate whether the persistent pulmonary hypertension following one month of reoxygenation after hypoxia would resolve upon prolonged reoxygenation, we assessed pulmonary hypertension after two months of reoxygenation. We found persistent pulmonary hypertension, as judged by RVSP and right ventricular hypertrophy, in both the EC-*Atm*⁻/⁻ and EC-*Bmpr2*⁻/⁻ mice two months after hypoxia was terminated (Supplementary Fig. 3f, g).

### Transcriptome analysis of gene expression in EC-*Atm*⁻/⁻ and EC-*Bmpr2*⁻/⁻ mice

To investigate the relationship between persistent DNA damage and pulmonary hypertension following reoxygenation, we isolated murine lung EC from EC-*Atm*⁻/⁻, EC-*Bmpr2*⁻/⁻ and control mice (Fig. 4a). Compared to control mice, 547 genes were commonly downregulated (Fig. 4b) and 512 genes were commonly upregulated (Supplementary Fig. 4a) in EC-*Atm*⁻/⁻ and EC-*Bmpr2*⁻/⁻ mice. Downregulated genes were related to EC differentiation and development (Fig. 4b) while upregulated genes were associated with leukocyte migration and immune response (Supplementary Fig. 4a). The top 50 genes that are significantly up or downregulated compared to control mice in either EC-*Atm*⁻/⁻ or *Bmpr2*⁻/⁻ are shown in Supplementary Table 2.

Among the commonly downregulated genes in EC-*Atm*⁻/⁻ and EC-*Bmpr2*⁻/⁻ mice were several transcription factors including those important in EC homeostasis, such as *Klf4, Klf2* and *Foxf1*. FOXF1 is a transcription factor predicted by motif analysis to regulate *KLFs and ERG*, and FOXF1 is implicated in DNA repair[28] as it interacts with the Fanconi anemia protein to promote cell survival and genome maintenance after DNA damage[28]. FOXF1 is also important in lung vascular formation during development[29] and is known to be a transcription target of P53[30]. Heterozygous mutations of *FOXF1* are causal in alveolar capillary dysplasia, a fatal disease of infancy with misaligned pulmonary arteries and veins and extensive occlusive changes in pulmonary arteries similar to those observed in PAH[31]. A number of genes that are

similarly reduced in EC-*Atm*⁻/⁻ and EC-*Bmpr2*⁻/⁻ mice including *Vegfr2 (Kdr), S1pr1, Angpt1*, and *Acvrl1*[32–34] are regulated by FOXF1 (Fig. 4c, d). Furthermore, genes in which mutations are described in PAH including *Bmpr2, Acvrl1, Eng, Vegfr2* and *Smad9*[35,36] are downregulated in both EC-*Atm*⁻/⁻ and EC-*Bmpr2*⁻/⁻ lung EC. Others that are reduced in expression in these two transgenic lines include *Cldn5* and *Cdh5*, cell surface molecules associated with endothelial identify and angiogenesis[37,38] and *Trp53inp2*, a gene downstream of P53 (Fig. 4c, d). Of 493 FOXF1 mediated target genes that were also DNA damage repair genes (GO 0006281) obtained from databases[39], 11 were differentially expressed in EC-*Bmpr2*⁻/⁻ lung EC vs. control lung EC (*Cul4a, Rrm2b, Hspa1a, Babam1, Rad21, Fancc, Morf4l1, Axin2, Smarca2, Rad51ap1 and Cep164*). This represented 11/14 total DNA repair genes that were differentially expressed. Interestingly, 5 of the 11 DNA repair genes that are targets of *Foxf1* are downregulated in lung EC of both EC-*Atm*⁻/⁻ and EC-*Bmpr2*⁻/⁻ mice (*Fancc, Smarca22, Rad21, Rrm2b and Cep164*) (Fig. 4c, d).

Genes upregulated in EC-*Atm*⁻/⁻ and EC-*Bmpr2*⁻/⁻ lung EC are related to inflammation wound healing and endothelial-mesenchymal transition. These include *Thbs1, Fn1 and Fbln2* (Supplementary Fig. 4). Transcription factors were not represented in the top 50 upregulated genes (Supplementary Table 2).

We confirmed that mRNA expression of *Foxf1* was significantly decreased in EC-*Atm*⁻/⁻ and EC-*Bmpr2*⁻/⁻ mice after reoxygenation compared with control mice (Fig. 4e), and FOXF1 protein was also decreased (Fig. 4f). We then also verified the reduction in proteins corresponding to some of the transcripts that were decreased in EC-*Atm*⁻/⁻ and EC-*Bmpr2*⁻/⁻ lung EC including VEGFR2 and CLDN5 or that are involved in DNA damage-repair such as ATM and P53 (Fig. 4f). These findings are consistent with the importance of ATM and P53, both in the sensing and repair of DNA damage and in the recovery of angiogenesis genes important in the restoration of small arteries.

### Single-cell transcriptome analysis in EC types in EC-*Bmpr2*⁻/⁻ mice

Recent single-cell RNA sequencing data examining tissue-specific vascular endothelial heterogeneity in adult mice suggest that genes specifically expressed in lung EC are regulated by *Foxf1*[40], further supporting its role as a lung EC-specific transcription factor. In humans, *FOXF1* is expressed in PAEC and aerocytes, recently discovered capillary EC that specialize in gas exchange and the trafficking of leukocytes[41], as well as general lung capillary EC[42]. *Foxf1* has a similar profile of expression in murine EC (https://tabula-muris.ds.czbiohub.org)[43], (Supplementary Fig. 5). To characterize the expression of *Foxf1* and its downstream targets in different EC subtypes in response to *Bmpr2* deletion, we compared lung EC single-cell transcriptomes from EC-*Bmpr2*⁻/⁻ and control mice following reoxygenation. In this model, tamoxifen injection induced tdTomato in endothelial cells. Accordingly, we examined tdTomato⁺ cells and identified five EC subtypes (lymphatic EC, arterial EC, venous EC, general capillary EC, and aerocytes) on the basis of previously reported mouse EC subpopulation markers[41,44] (Fig. 5a–c). As expected, *Bmpr2* was expressed in all EC subtypes, and was strongly and significantly down-regulated in EC-*Bmpr2*⁻/⁻ versus control mice (Fig. 5d–f). *Foxf1* and downstream genes involved in angiogenesis, *Vegfr2 (Kdr)* and *Clnd5*, were also expressed in all EC subtypes (with the exception of *Foxf1* in lymphatic EC) (Fig. 5e, f). *Foxf1* and its downstream target *Vegfr2 (Kdr)* was decreased in EC-*Bmpr2*⁻/⁻ mice in all clusters, while *Cldn5* was specifically reduced in aerocytes and gCap (Fig. 5e, f).

### Gain and loss of FOXF1 in PAEC relate to DNA damage and angiogenesis

We detected FOXF1 in the nuclei of PAEC, and the intensity in EC-*Atm*⁻/⁻ mice and EC-*Bmpr2*⁻/⁻ mice was decreased relative to that in control mice in normoxia (Supplementary Fig. 6a and Fig. 6a). FOXF1 intensity in EC-*Bmpr2*⁻/⁻ mice was decreased to the same level as control mice in

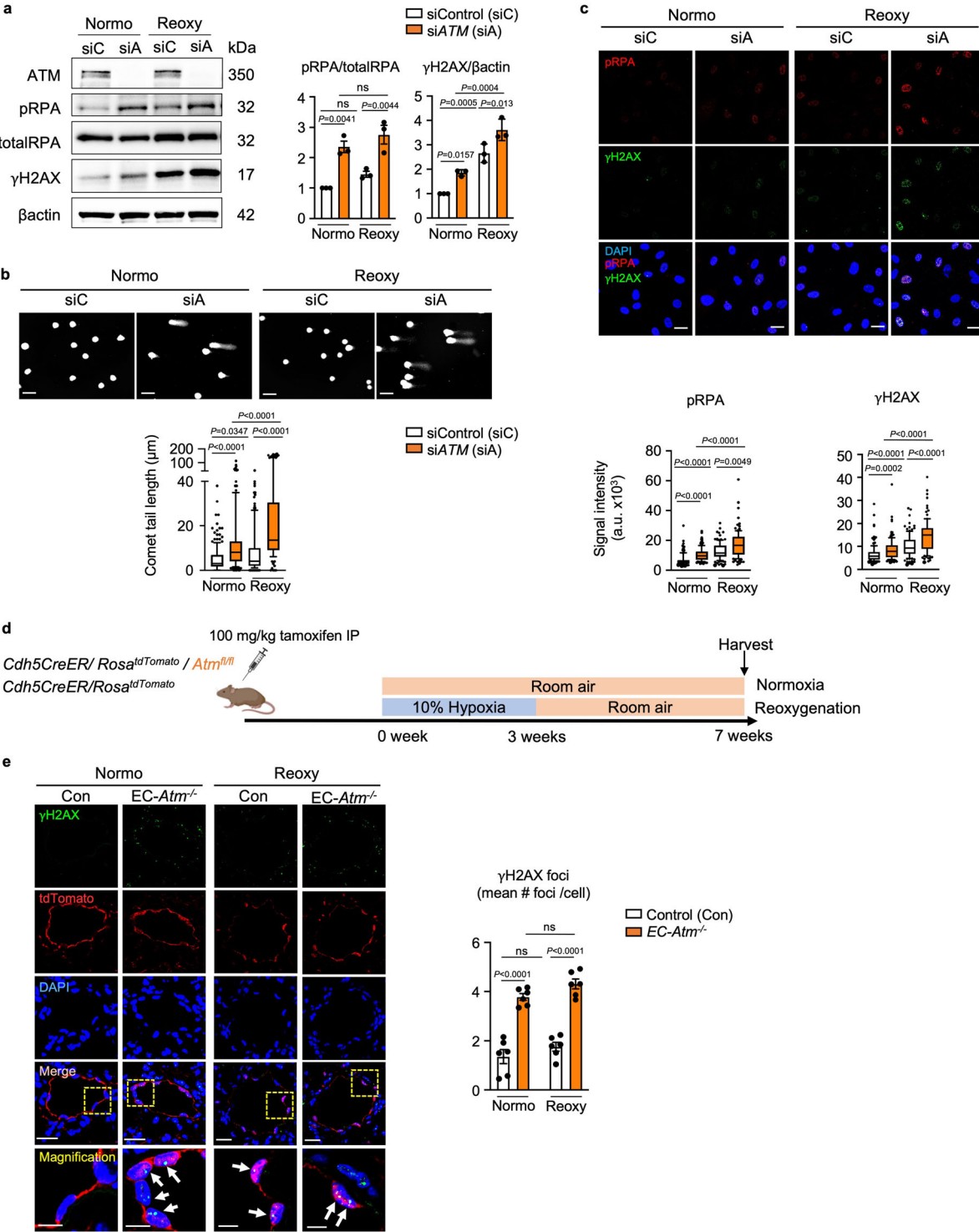

**Fig. 2 | Loss of *ATM* induces unrepaired DNA damage in normoxia and after reoxygenation in human PAEC and in EC-*Atm*⁻/⁻ mice. a–c** Commercially available healthy donor PAEC were cultured under hypoxia (0.5% O₂) for 48 h followed by room air for 24 h (reoxy), or in room air for 72 h (normo). **a** Representative immunoblot of DNA damage markers (γH2AX and phosphorylated RPA) in human PAEC transfected with scrambled control siRNA (siC) or siRNA targeting *ATM* (siA). *n* = 3 individual experiments. **b** Comet assay shows DNA damage as reflected in the comet tail lengths in ATM-depleted cells in normoxia or reoxygenation. Scale bar, 20 μm. Cells *n* = 186, 154, 139, 124 for siC normoxia, siA normoxia, siC Reoxy, siA reoxy respectively. **c** Immunohistochemistry of the DNA damage markers in human PAEC transfected with Control or *ATM* siRNA. Scale bars, 20 μm. Cells *n* = 130, 133, 117, 102 for siC normoxia, siA normoxia, siC Reoxy, siA reoxy respectively. **d** *Animal measurements* - experimental design: Mice with EC-specific deletion of *Atm* (EC-

*Atm*⁻/⁻) were created using the strategy described for Fig. 1, and in the "Methods". EC-*Atm*⁻/⁻ or control mice were exposed to hypoxia (10% oxygen) for 3 weeks followed by 4 weeks of room air (reoxy), or maintained in room air for 7 weeks (normo). Schema created with BioRender.com. **e** γH2AX immunofluorescence foci in PAEC in control and EC-*Atm*⁻/⁻ mice (arrowheads) were quantified in n = 6 mice. Scale bars, 20 μm. The bottom panels show a magnified merged image of the area delineated by the dotted line. Scale bars, 5 μm. In (**a** and **d**), bars represent mean ± S.E.M. *P* values determined by 2-way ANOVA with Holm-Sidak posthoc test. ns, not significant. In (**b** and **c**) The Bounds of boxes show the 25th and 75th percentiles, the whisker showed to the 10th to 90th percentiles and the centre in the box shows the median. Three independent experiments was performed. *P* values determined by Kruskal-Wallis ANOVA test with Dunn's test Source data are provided as a Source Data file.

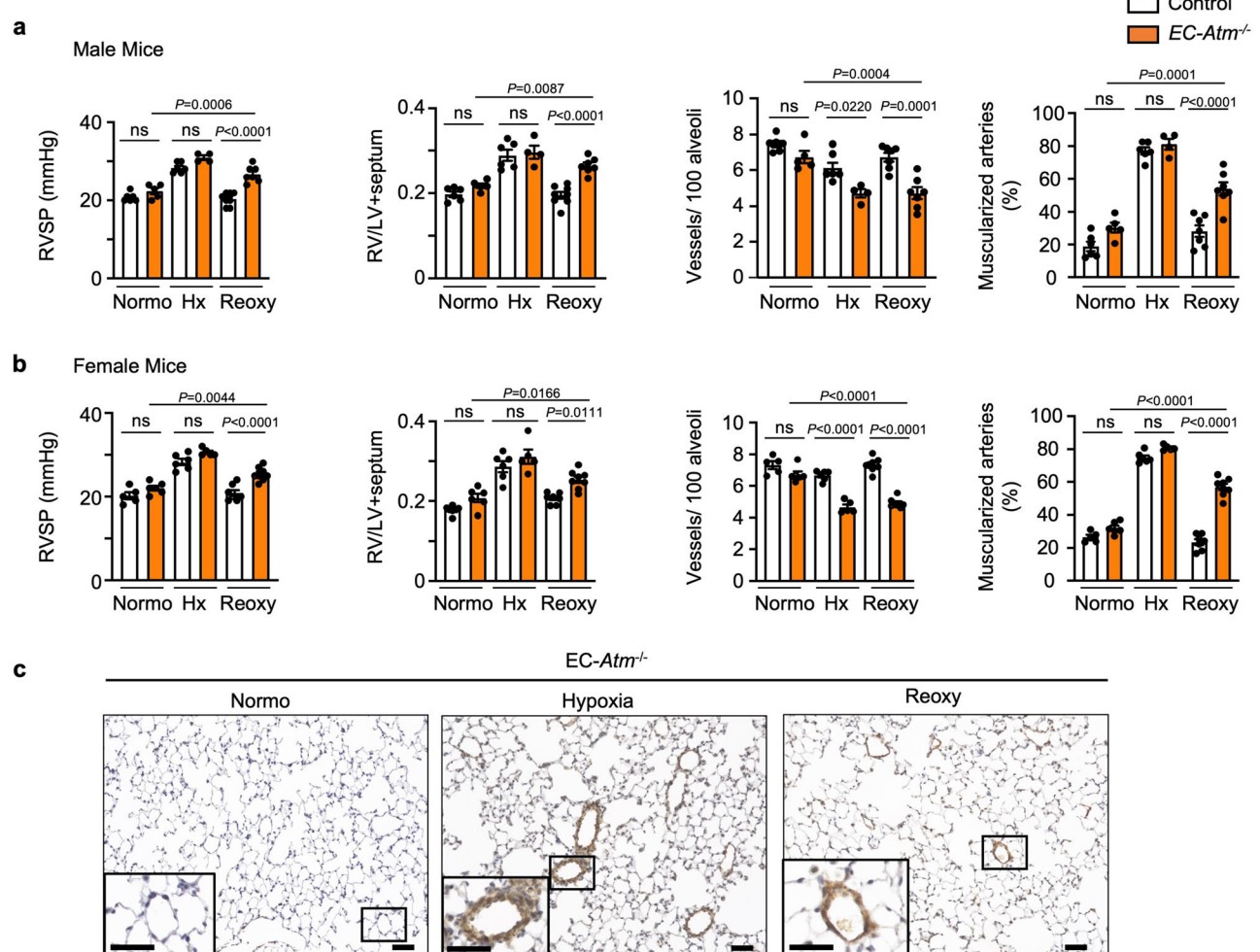

**Fig. 3 | EC specific *Atm* deletion in mice induces persistent PAH after reoxygenation.** EC-*Atm*[-/-] and control mice were subjected to (i) 7 weeks of room air (Normo), or (ii) 3 weeks of hypoxia (10% $O_2$, Hx) only, or (iii) 3 weeks of hypoxia (10% $O_2$), followed by 4 weeks of reoxygenation in normoxia (Reoxy). Right ventricular (RV) systolic pressure (RVSP), RV hypertrophy (weight of RV/left ventricle and septum, RV/LV + S), number of vessels per 100 alveoli at alveolar wall and duct level, and the percentage of fully or partially muscularized pulmonary arteries in (**a**) male mice (Control: Normo $n = 6$, Hx $n = 6$, Reoxy $n = 8$; EC-*Atm*[-/-]: Normo $n = 5$, Hx $n = 4$, Reoxy $n = 7$) and (**b**) female mice (Control: Normo $n = 5$, Hx $n = 6$, Reoxy $n = 7$; EC-*Atm*[-/-]: Normo $n = 6$, Hx $n = 5$, Reoxy $n = 8$). Each data point represents a mouse. Bars represent mean ± S.E.M. P values determined by 2-way ANOVA with Holm-Sidak posthoc test. ns, not significant. **c** Representative images of αSMA staining of lung section in EC-*Atm*[-/-] mice. Scale bars, 20 μm. Inserts in the lower left corners show a magnified image of vessels delineated by the black lines (Scale bars, 20 μm). Source data are provided as a Source Data file.

hypoxia (Supplementary Fig. 6b). However, following reoxygenation, FOXF1 intensity was reduced relative to that of control mice (Fig. 6a). Consistent with FOXF1 intensity, lung EC *Foxf1* mRNA levels were decreased in EC-*Bmpr2*[-/-] and EC-*Atm*[-/-] mice under normoxia and reoxygenation (Fig. 4e and Supplementary Fig. 6c). In tissue sections of human pulmonary arteries, FOXF1 intensity was reduced in vWF-positive EC in PAH obstructive lesions (Fig. 6b) and in some vWF-positive EC in plexiform lesions (Supplementary Fig. 6d), compared to control lung PAEC. Interestingly, FOXF1 intensity in vWF-positive EC was decreased in PAH patients with or without a *BMPR2* mutation (Fig. 6b), as were the mRNA levels of *FOXF1* and *ATM* in PAH PAEC (Supplementary Fig. 6e). This is consistent with a reduction in *BMPR2* in PAEC even in PAH patients without a BMPR2 mutation compared to controls (Supplementary Fig. 6e)[4].

Next, we reduced *FOXF1* in human PAEC using *FOXF1*-targeting siRNA (si*FOXF1*) vs. control siRNA and assayed selective targets of FOXF1, including *VEGFR2* and *CLDN5*. We also monitored the expression of *TP53* and *ATM*, and although they are upstream of FOXF1, we found them to be reduced, suggesting a negative feedback loop (Fig. 6c). Furthermore, loss of *FOXF1* in control PAEC significantly

reduced angiogenesis as assayed by a decrease in the number of tubes formed in Matrigel (Fig. 6d). Consistent with these findings, there was a reduction in migration of PAEC with loss of FOXF1 as judged by a wound healing scratch assay (Fig. 6e). Conversely, when FOXF1 was overexpressed in commercially available human PAEC the levels of VEGFR2, CLDN5, P53 and ATM were upregulated (Supplementary Fig. 6f).

Most relevant, when FOXF1 was transfected in PAEC of three PAH patients, expression levels of selected transcripts (*VEGFR2, CLDN5, TP53, and ATM*) were increased (Fig. 6f) This is consistent with restoration of angiogenesis, assessed by the Matrigel tube formation assay (Fig. 6g and Supplementary Fig. 6g), as well as PAEC migration, determined by the scratch wound assay (Fig. 6h and Supplementary Fig. 6h) and repair of DNA damage, judged by comet assay (Fig. 6i and Supplementary Fig. 6i and j).

### *Foxf1* delivery by adeno-associated virus (AAV) restores angiogenesis and DNA repair and reverses persistent PH in mice

To investigate whether the delivery of *Foxf1* could reverse persistent PAH, we used a tagged AAV vector, AAV2-ESGHGYF, that specifically

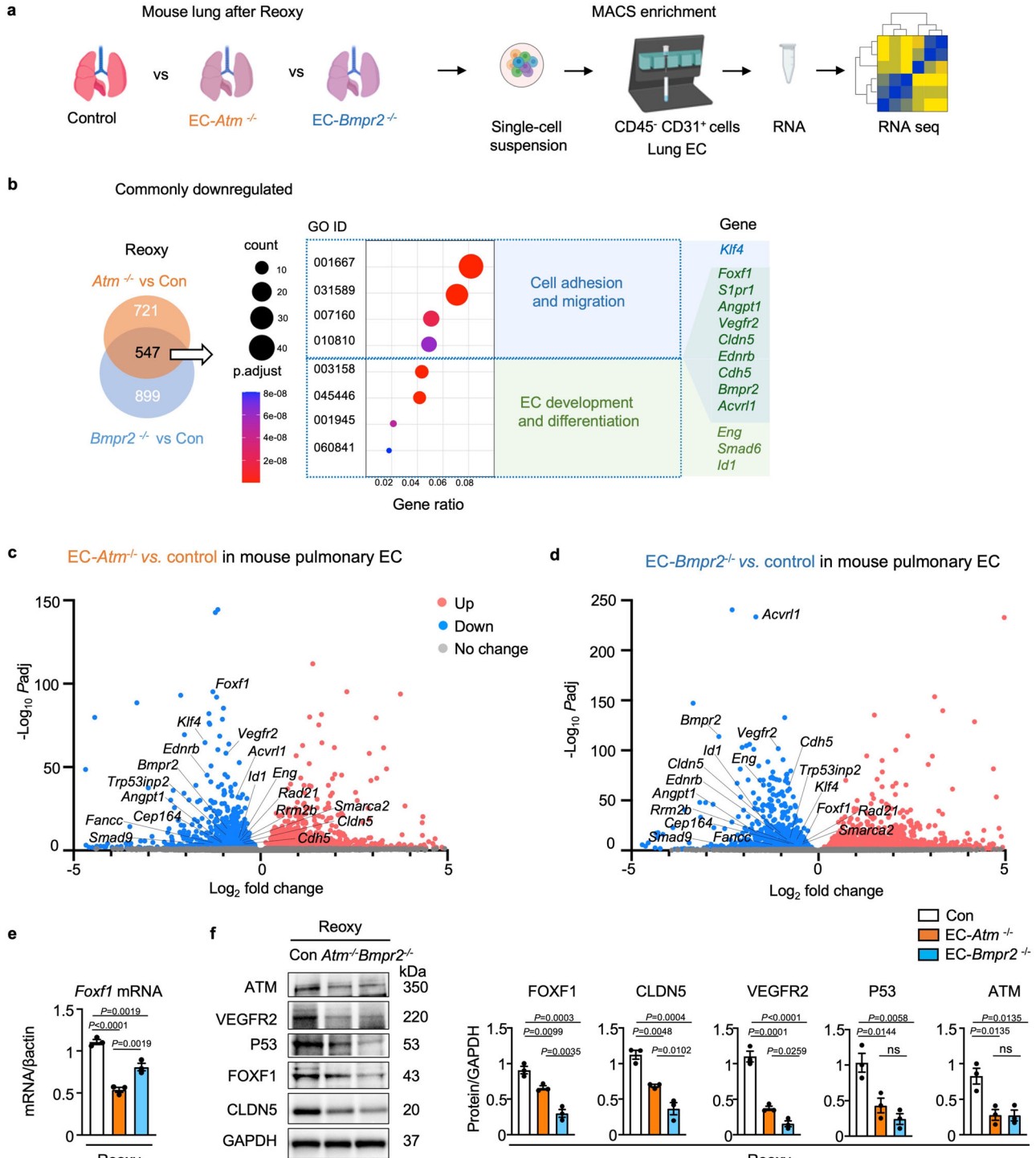

**Fig. 4 | Transcriptome analysis reveals that angiogenesis pathways are down-regulated in *Bmpr2*- and *Atm*-deleted murine PAEC. a** Scheme of study design. The lungs of tamoxifen-inducible EC-*Atm*-/-, EC-*Bmpr2*-/- and control mice (*n* = 5) after reoxygenation were harvested, enzymatically digested to obtain single-cell suspensions, and pulmonary endothelial cells (CD45- CD31+ cells) were collected using magnetic beads. Total RNA of pulmonary EC from *n* = 5 mice was used for bulk RNA sequencing. Schema created with BioRender.com. **b** Venn diagram showing the number of genes downregulated following reoxygenation in lung EC of EC-*Atm*-/- or EC-*Bmpr2*-/- mice, compared to control mice [adjusted *P*-value (*P*adj) <0.05]. Gene Ontology (GO) analysis was performed for 547 commonly down-regulated genes to reveal the top GO pathways. *P*-values were adjusted using the Benjamini–Hochberg method for multiple comparisons by one-sided Fisher's exact test. **c, d** Volcano plots displaying DEGs (differentially expressed genes) following

reoxygenation in EC-*Atm*-/- or EC-*Bmpr2*-/- compared to control mice (c, d respectively). Angiogenesis genes (*Foxf1, Vegfr2, Acvrl1, S1pr1, Eng, Bmpr2, Id1, Klf4, Cdh5, Cldn5, Ednrb*), *Trp53inp2*, a gene downstream of p53, and *Foxf1* targeting as well as DNA repair genes (*Fancc, Smarca2, Rad21, Rrm2b, Cep164*) are indicated on the plots. 2-sided Wald test was performed. *P*-values were adjusted to get *P*adj using the Benjamini-Hochberg method. **e** *Foxf1* expression in lung EC of EC-*Atm*-/-, EC-*Bmpr2*-/- and control mice following reoxygenation. *n* = 3 biological replicates. **f** Immunoblot and densitometry of FOXF1, CLDN5, VEGFR2, P53 and ATM in pulmonary EC of EC-*Atm*-/-, EC-*Bmpr2*-/- and control mice after reoxygenation. *n* = 3 biological replicates. **e, f** Bars represent mean ± S.E.M. *P* values determined by one-way ANOVA with Holm-Sidak posthoc test. ns, not significant. Source data are provided as a Source Data file.

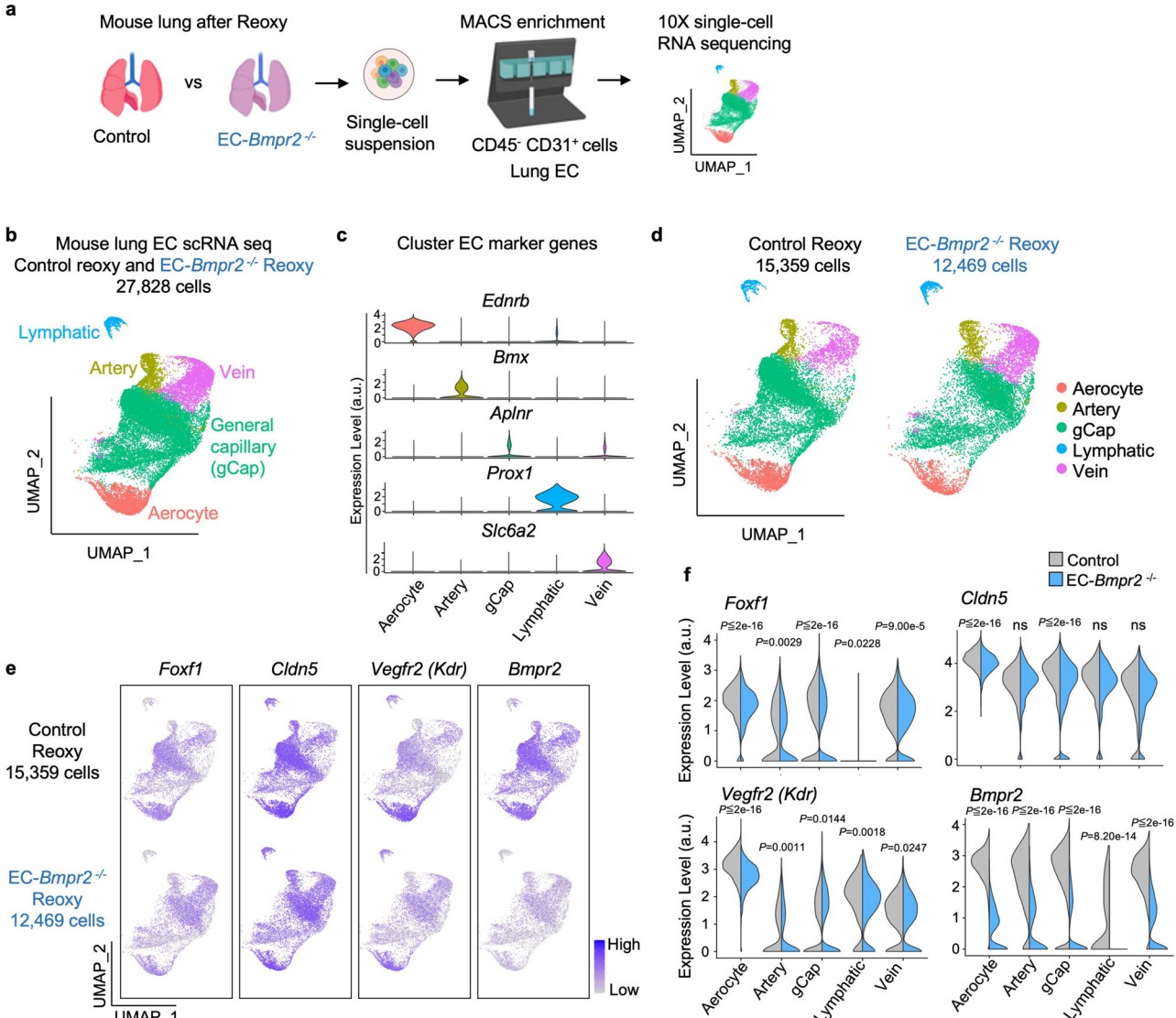

**Fig. 5 | Single-cell RNA seq identifies decreased angiogenesis genes in EC sub-populations of *Bmpr2*-deleted mice following reoxygenation. a** Scheme of study design. The right lungs of tamoxifen inducible EC-*Bmpr2*⁻/⁻ (*n* = 3 mice) or control mice (*n* = 4 mice) after reoxygenation were harvested, pooled, and enzymatically digested to obtain single-cell suspensions. Pulmonary endothelial cells (CD45⁻ CD31⁺ cells) were collected using magnetic beads and analyzed by 10X single-cell RNA sequencing. Schema created with BioRender.com. **b** Uniform Manifold Approximation and Projection (UMAP) of tdTomato⁺ traced lung EC in EC-*Bmpr2*⁻/⁻ and control mice following reoxygenation (*n* = 27,828 cells). Pulmonary EC (CD45⁻CD31⁺ cells) were selected using magnetic beads and single-cell RNA seq performed on tdTomato-positive cells (tdTomato gene expression > 0). EC clusters were identified using EC subpopulation markers: *Ednrb* for aerocytes; *Bmx for* artery EC; *Aplnr* for general capillary (gCap); *Prox1* for Lymphatic EC; *Slc6a2* for Vein EC. Numbers of cells in each subtype: Aerocytes, *n* = 3574 cells; Artery EC, *n* = 1798

cells; gCap, *n* = 17,920 cells; Lymphatic EC, *n* = 724 cells; Vein EC, *n* = 3812 cells. **c** Violin plots of the expression of the EC subpopulation markers *Ednrb, Bmx, Aplnr, Prox1* and *Slc6a2*. a.u., arbitrary units. **d** UMAP in (a), separated to control (left; *n* = 15,359 cells) and EC-*Bmpr2*⁻/⁻ (right; *n* = 12,469 cells) mice. Numbers of cells in control mice: Aerocytes, *n* = 2232 cells; artery EC, *n* = 882 cells; gCap, *n* = 10,616 cells; Lymphatic EC, *n* = 324 cells; Vein EC, *n* = 1305 cells. Numbers of cells in EC-*Bmpr2*⁻/⁻ mice: Aerocytes, *n* = 1342 cells; artery EC, *n* = 916 cells; gCap, *n* = 7304 cells; Lymphatic EC, *n* = 400 cells; Vein EC, *n* = 2507 cells. **e**. UMAP of angiogenesis genes colored by expression levels in EC-*Bmpr2*⁻/⁻ or control mice following reoxygenation. Color scale: Purple, high expression; white, low expression. **f** Violin plots showing expression of angiogenesis genes in each pulmonary EC subpopulation of EC-*Bmpr2*⁻/⁻ or control mice following reoxygenation. a.u., arbitrary units. *P* values are determined by 2-sided Wilcoxon rank sum test. ns, not significant. Source data are provided as a Source Data file.

targets an increase in expression of *Foxf1* to the pulmonary vascular endothelium[23]. AAV-ESGHGYF carrying the luciferase gene served as a control vector. Mice were exposed to hypoxia (10% O₂) for three weeks, followed by administration of AAV via tail vein injection, and returned to room air for 4 weeks (reoxygenation). To ascertain selective delivery to the lungs, luciferase activity was examined one and four weeks later (see experimental schema in Fig. 7a). Luciferase activity was exclusively detected in the lungs (Fig. 7b), and immunostaining indicated colocalization with the vascular endothelial marker MECA in tdTomato-positive cells (Fig. 7c). Flow cytometry

analysis of whole lung from control mice treated with AAV-luciferase showed that luciferase was not expressed in the tdTomato-negative population (0.89%) when compared to tdTomato-positive EC, where luciferase-positive cells represented 24.7% of the total lung digest (Supplementary Fig. 7a, b). In EC-*Bmpr2*⁻/⁻ mice after 4 weeks of reoxygenation, AAV-*Foxf1* reversed persistent pulmonary hypertension judged by normalization of right ventricular systolic pressure and right ventricular hypertrophy (Fig. 7d), pulmonary artery acceleration time/ejection time (PAAT/ET) (Supplementary Fig. 7c), the number of arteries per 100 alveoli at alveolar wall and duct level, and

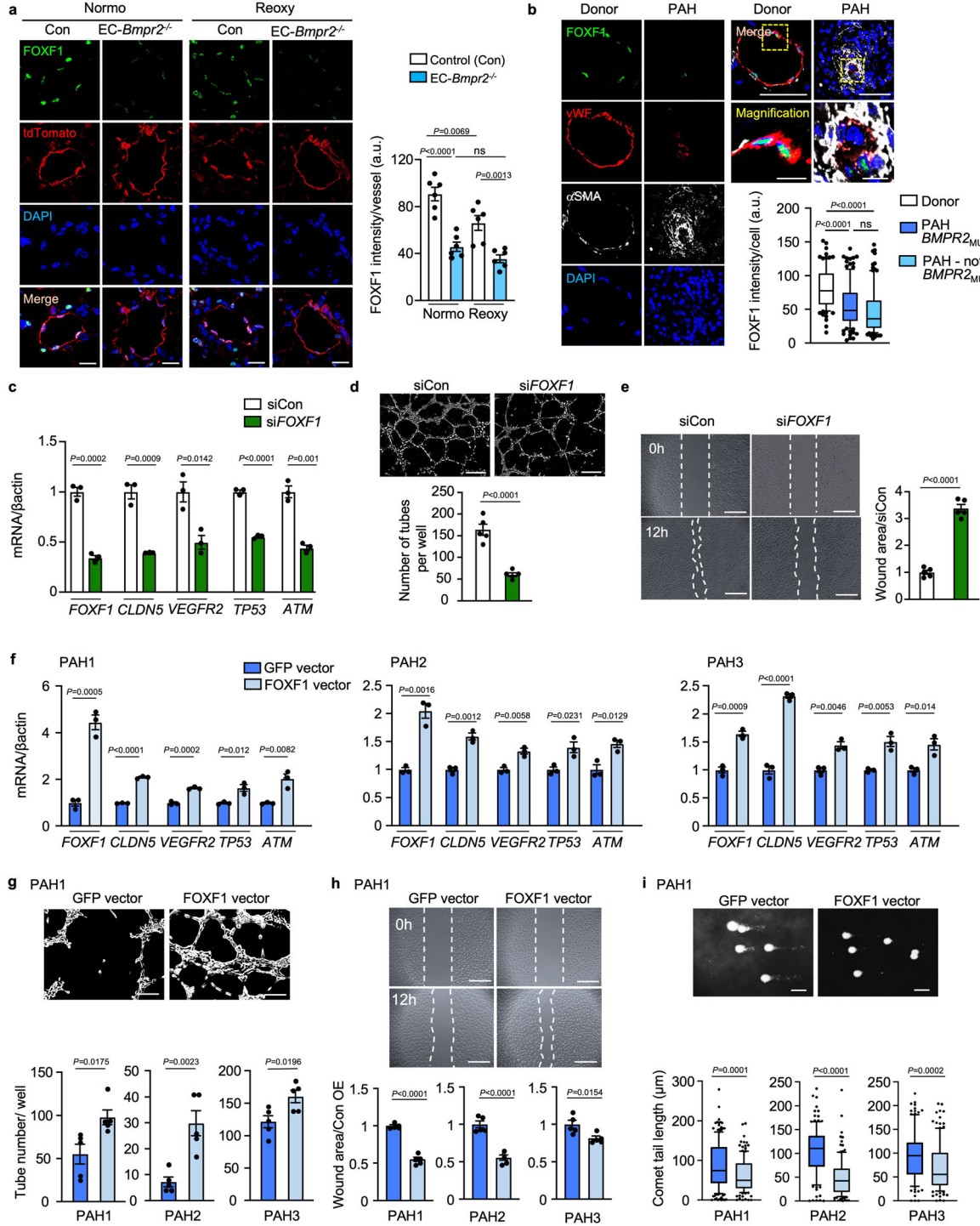

**Fig. 6 | FOXF1 in PAEC is related to angiogenesis and DNA damage repair.**
**a** FOXF1 immunohistochemistry of pulmonary artery (PA) in EC-*Bmpr2*$^{-/-}$ mice under normoxia (Normo) or reoxygenation (Reoxy). FOXF1 intensity quantified in 5 vessels/mouse (*n* = 6 mice/group), Scale bar, 20 μm. **b** Representative images of an occlusive PAH lesion and a normal donor PA, with quantification of FOXF1 intensity in PAEC (von Willebrand factor (vWF)-positive cells), in PAH BMPR2$_{MUT}$, PAH-non BMPR2$_{MUT}$ and healthy donors (*n* = 4/group). αSMA indicates smooth muscle cells. Scale bar, 50 μm. Bottom panels show magnified merged image of the area delineated by the dotted line. Scale bars, 10 μm. Five arteries analyzed in each patient with *n* = 96, 114, 109 cells for Donor, PAH-BMPR2$_{MUT}$ and PAH-not MPR2$_{MUT}$, respectively. **c–e** Angiogenesis and DNA damage following reduced *FOXF1*: Commercially available human PAEC treated with siRNA targeting *FOXF1* (si*FOXF1*) vs. control siRNA (siCon). **c** mRNA levels. *n* = 3 replicates. **d** Representative images of tube formation assay, with quantitative analysis 3 h after seeding. *n* = 5 replicates. Scale bar, 200 μm. **e** Wound closure (scratch) assay by cell migration.

Representative images at 0 and 12 h, and quantified ratio of the area of the scratch in si*FOXF1* relative to siCon at 12 h. *n* = 5 replicates. Scale bar, 450 μm. **f–i** Angiogenesis and DNA damage after restoration of FOXF1 expression. PAH PAEC transfected with lentivirus vector carrying the *FOXF1* gene or GFP as control. **f** mRNA of angiogenesis and DNA damage response genes (*n* = 3 replicates) in cells of PAH patients PAH1, PAH2 and PAH3. **g, h** Tube formation and scratch assays, in PAEC of PAH patients (*n* = 5 replicates/patient). Scale bars, 200 μm in (g) 450 μm in (**h**). **i** Comet assay: DNA damage reflected by the comet tail length. Scale bar, 20 μm. PAH1, *n* = 173, 131; PAH2, *n* = 125, 129; PAH3; *n* = 116, 123 cells for GFP-vector, *FOXF1*-vector, respectively. Supplementary Fig. 6 shows the images for tube formation, scratch and comet assays for patients PAH2, PAH3. **a, c–h** Bars represent mean ± S.E.M; (**b, i**), Box bounds = 25th and 75th percentiles, whiskers 10th to 90th percentiles, and box centre = the median. *P* values were determined by one-way ANOVA with Holm-Sidak posthoc test (**a**), 2-sided Mann–Whitney U test (**b, i**), or unpaired 2-sided t-test (**c–h**). ns, not significant. Source data are provided as a Source Data file.

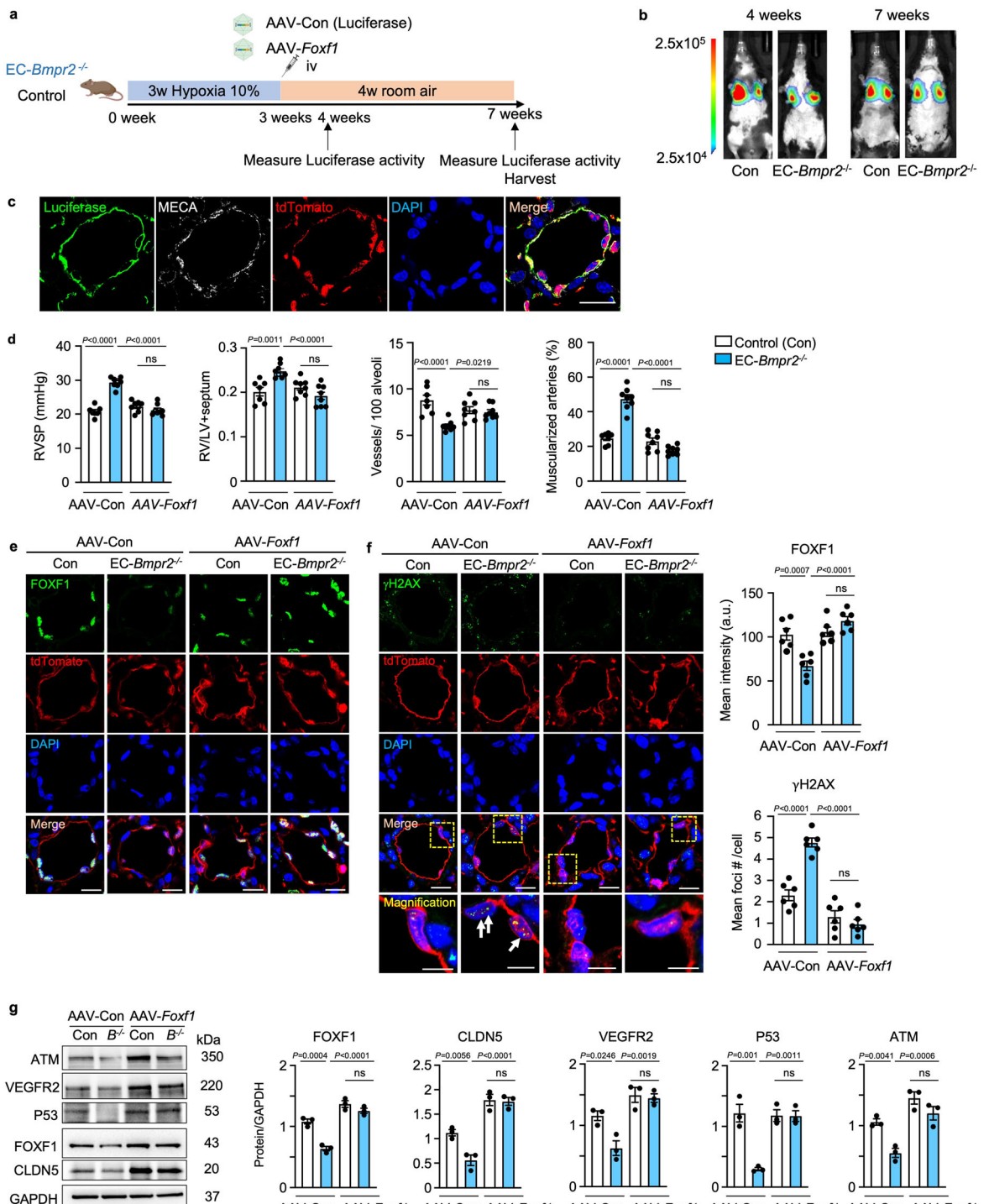

**Fig. 7 | Delivering *Foxf1* to pulmonary endothelial cells of EC-*Bmpr2*⁻/⁻ mice restores angiogenesis and reverses persistent pulmonary hypertension.**
**a** Experimental design: AAV carrying *Foxf1* (AAV-*Foxf1*) or luciferase (AAV-Con) as control were administered via tail vein injection to EC-*Bmpr2*⁻/⁻ or control mice following 3 weeks in hypoxia (10% O₂). The mice were then returned to room air for 4 weeks. Schema created with BioRender.com. **b** In vivo luciferase activity was measured in EC-*Bmpr2*⁻/⁻ and control mice by LagoX in vivo imaging tool as described in the "Methods", at 4 and 7 weeks (1 and 4 weeks after AAV injection). **c** Immunohistochemistry showing luciferase expression colocalized with MECA, a pan-EC marker, and with tdTomato-positive cells in a pulmonary artery of EC-*Bmpr2*⁻/⁻ mice treated with AAV-luciferase for four weeks. Three lung sections were analyzed in *n* = 2 mice/group, with approximately 20 vessels observed. Scale bar, 20 μm. **d** Right ventricular systolic pressure (RVSP), right ventricular hypertrophy, the weight of right ventricle relative to the left ventricle+septum (RV/LV + S), number of vessels per 100 alveoli and the percent of fully or partially muscularized

pulmonary arteries in male EC-*Bmpr2*⁻/⁻ and control mice treated with AAV-Con or AAV-*Foxf1*. Control mice with AAV-Con, *n* = 7; EC-*Bmpr2*⁻/⁻ mice with AAV-Con, control mice with AAV-*Foxf1*, and EC-*Bmpr2*⁻/⁻ mice with AAV-*Foxf1*, *n* = 8. **e, f** FOXF1 (**e**) and γH2AX (**f**) immunohistochemistry of pulmonary arteries of EC-*Bmpr2*⁻/⁻ and control mice treated with AAV-Con or AAV-*Foxf1* following reoxygenation. The average FOXF1 intensity and number of γH2AX foci were quantified in 5 vessels per mouse, in *n* = 4 mice per group. In EC-*Bmpr2*⁻/⁻ mice treated with AAV-*Foxf1*, FOXF1 levels were restored to those of control mice, and γH2AX immunofluorescence foci number decreased to the level in control mice. Scale bar, 20 μm. The bottom panels in (**f**) show a magnified merged image of the area delineated by the dotted line (**f**). Scale bar, 5 μm. **g** Immunoblot and densitometry of FOXF1, CLDN5, VEGFR2, P53 and ATM in pulmonary EC in EC-*Bmpr2*⁻/⁻ and control mice after AAV-*Foxf1* or AAV-Con treatment. *n* = 3 biological replicates. **d–g** Bars represent mean ± S.E.M. *P* values determined by 2-way ANOVA with Holm-Sidak posthoc test. ns, not significant. Source data are provided as a Source Data file.

the percentage of muscularized arteries at these levels when compared with AAV-luciferase transfected control mice (Fig. 7d). The increase in small vessels may in part have been responsible for the reduction in the percent of vessels that were muscularized. Immunostaining in pulmonary arteries in lung tissue sections showed that FOXF1 returned to control levels in EC-*Bmpr2*[-/-] mice following AAV-*Foxf1* administration (Fig. 7e). Furthermore, the number of γH2AX foci decreased after AAV-*Foxf1* injection to EC-*Bmpr2*[-/-] mice, suggesting that DNA damage was repaired (Fig. 7f). In addition to elevating the expression of FOXF1 to control levels, *Foxf1* delivery increased angiogenesis genes CLDN5 and VEGFR2, and restored DNA damage repair genes P53 and ATM (Fig. 7g).

## Discussion

Pulmonary arterial hypertension is caused by occlusive changes in peripheral pulmonary arteries that are associated with DNA damage in PAEC and PASMC[5-7]. Here we used reoxygenation after hypoxia in EC-*Atm*[-/-] and EC-*Bmpr2*[-/-] mice to find a link between unrepaired DNA damage and persistent pulmonary hypertension. We found that reduced FOXF1 and target genes were related to impaired angiogenesis and DNA repair in mice and in human PAEC from PAH patients. Loss of FOXF1 in control human PAEC resulted in DNA damage and impaired angiogenesis, and gain of FOXF1 in PAH PAEC reversed these abnormalities. In EC-*Bmpr2*[-/-] mice, directing *Foxf1* to lung EC during reoxygenation restored normal peripheral pulmonary arteries and allowed for the regression of pulmonary hypertension (Fig. 8).

We found that DNA damage is increased by loss of BMPR2 in normoxia. We have previously shown that decreased BMPR2 results in extensive mitochondrial fission and an increase in ROS[18], a major cause of DNA damage, and activation of γH2AX. During hypoxic exposure, both control cells and those with loss of BMPR2 showed a similar reduction in p53[18], and an increase in γH2AX. Based upon previous studies, the mechanism causing hypoxia-induced DNA damage is replication arrest[45] and phosphorylation of γH2AX in an ATR (Ataxia telangiectasia and Rad3 related)-dependent manner[46]. In response to reoxygenation[47] when *BMPR2*[18] is decreased, P53 phosphorylation is reduced in PAEC, thereby resulting in unrepaired DNA damage. This explains why those who experience reoxygenation after hypoxia do not develop PAH unless they have associated *BMPR2* or other genetic abnormalities[48] and why genotoxic agents such as amphetamine also

require a second hit to develop PAH[49,50]. Inflammation[51], chemotherapy[52] and radiation[53] are all genotoxic modifiers that could be working as second hits in patients genetically predisposed to PAH. We reduced *ATM* in human PAEC to investigate whether unrepaired DNA damage could be causally linked to PAH. We found that loss of ATM induces unrepaired DNA damage judged by an increase in γH2AX during normoxia despite activation of alternative DNA repair pathways such as ATR and DNA-PK[26,54] (Supplementary Fig. 2c). With hypoxia and reoxygenation, we speculate that loss of ATM results in activation of the same pathway as loss of BMPR2. In the EC-*Bmpr2*[-/-] mouse, unrepaired DNA damage is induced by loss of *BMPR2*, and enhanced by oxidative stress caused by reoxygenation. Interestingly, *Vegfr2* expression in lung EC was reduced in this model, similar to the Sugen/Hypoxia rat model, the most common preclinical model of Group 1 PAH, where Sugen suppresses vascular endothelial growth factor receptors (VEGFR2; FLK1/KDR) and causes endothelial cell dysfunction.

To determine how unrepaired DNA damage could be linked to persistent pulmonary hypertension seen in EC-*Bmpr2*[-/-] mice, we exposed EC-*Atm*[-/-] mice to hypoxia and reoxygenation. We confirmed that they were similar in having no pulmonary hypertension during normoxia despite DNA damage, no hemodynamic or structural features more severe than control mice during hypoxia, but persistent pulmonary hypertension and unrepaired DNA damage after a month of reoxygenation. This indicates that in both EC-*Bmpr2*[-/-] and EC-*Atm*[-/-] mice, the ability to regenerate normal EC and to induce regression of muscularization of distal arteries may result from a similar mechanism tied to unrepaired DNA damage. Loss of *Bmpr2* or *Atm* result in impaired activation of P53, a protein necessary in DNA repair and in the induction of angiogenesis genes during reoxygenation[20,55-57]. The transcription of genes downstream of P53, such as apelin[20], is required for a normal endothelial phenotype responsible for suppressing proliferation of SMC[5] whereas release of cytokines[7] from abnormal endothelial cells can induce DNA damage in neighboring SMC (Fig. 1f).

Patients with ATM heterozygous mutations causing Ataxia-telangiectasia have genome instability and develop hematological malignancies such as leukemia and lymphoma, as well as immunodeficiency and neurodegeneration. Vascular lesions, including telangiectasias (dilated blood vessels) are common in these patients, but there are no reports of pulmonary arterial hypertension[58]. This

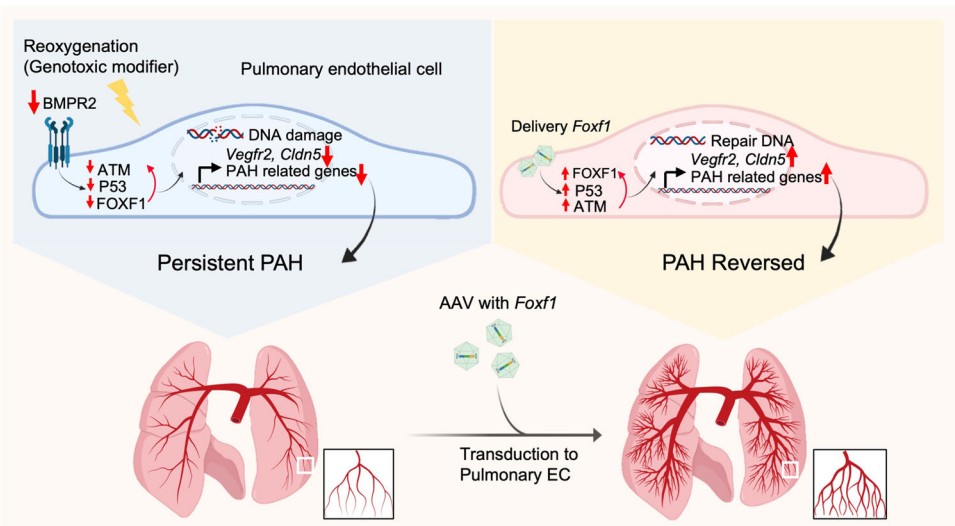

**Fig. 8 | Summary of Results.** Under oxidative stress reduced BMPR2 causes a reduction in FOXF1 as well as P53 and ATM, all required for DNA repair. FOXF1 is also required to transcribe angiogenesis genes to regenerate damaged endothelium. Thus reduced FOXF1 leads to PAH. *Foxf1* delivery by AAV in a mouse with *Bmpr2* deleted in endothelium is sufficient to restore P53 and ATM to repair DNA damage and transcribe angiogenesis genes required to reverse persistent PAH. Created with BioRender.com.

suggests that a different phenotype is evident with one remaining normal allele from birth, whereas a more severe reduction in ATM in the adult can cause PAH owing to impaired angiogenesis or an inability to regenerate microvessels in response to injury.

We performed transcriptomic analyzes of PAEC of EC-*Atm*[-/-] and EC-*Bmpr2*[-/-] mice to further characterize commonly downregulated genes and the transcription factor that could be regulating them independent of, or in addition to, loss of P53. A number of common angiogenesis genes were reduced, including *Vegfr2 (Kdr), Acvrl1, Eng* and *Smad9*, that are reported as causal for PAH[35,59]. Single-cell transcriptome analysis revealed that in EC-*Bmpr2*[-/-] mice, both *Foxf1* and its targets were downregulated in most EC subtypes. *Foxf1* was prominent among the transcription factors commonly downregulated in both EC-*Atm*[-/-] and EC-*Bmpr2*[-/-] mice, and it was one of the top five downregulated transcription factor genes. FOXF1 regulates both DNA repair[28,30] and angiogenesis[32–34] and is also required for BMP9/ACVRL1 signaling[34]. FOXF1 was decreased in lung EC of EC-*Atm*[-/-] and EC-*Bmpr2*[-/-] mice, especially after reoxygenation. As we have shown previously[18] and in Fig. 4f, P53 was reduced in EC-*Bmpr2*[-/-] or EC-*Atm*[-/-] mice after reoxygenation. Since *Foxf1* is regulated by P53[30], we speculate that the reduction of P53 during reoxygenation may be responsible for the decrease in FOXF1. The relatively selective expression of *Foxf1* in the pulmonary vascular endothelium[40,42] made it a particularly attractive target for an intervention that would increase its availability in PAH. As a further connection between *Foxf1* and DNA damage in EC-*Atm*[-/-] and EC-*Bmpr2*[-/-] mice, we found 5 DNA repair genes including *Fancc* and *Rad51ap1* that are direct targets of FOXF1, and other DNA repair genes that are differentially expressed that could be indirect targets. FANCC forms a complex with other FA proteins such as FANCM and activates downstream BRCA1 and RAD51 to repair DNA damage[60]. Given that BRCA1[14] and RAD51[15] are also downstream targets of BMPR2 signaling, decreased activity of RAD51 and BRCA1 is likely induced by decreased FA complex formation following loss of FANCC. This was reinforced by our studies in the EC-*Bmpr2*[-/-] mice where targeted expression of *Foxf1* using an AAV vector engineered to be delivered to lung EC prevented persistent pulmonary hypertension during reoxygenation after hypoxia. While treatment with Nutlin-3a, an activator of P53[20] that regulates *FOXF1* expression[30], was comparable in preventing persistent pulmonary hypertension in the EC-*Bmpr2*[-/-] mice after reoxygenation, activation of P53 with Nutlin-3a or other small molecules that inhibit MDM2 in cancer patients is associated with adverse systemic effects such as gastrointestinal toxicity, neutropenia and thrombocytopenia[61,62]. Other studies showed that BET inhibitors that prevent DNA repair and induce apoptosis of SMC[63] or senolytic drugs[64] may be considered as PAH therapy. However, a recent report showed worsening of experimental pulmonary hypertension with senolytics by removing EC required for regeneration of a damaged endothelium[65]. It is interesting that there is a report of PAH patients with *FOXF1* mutations that do not have alveolar capillary dysplasia (ACD) with misalignment of pulmonary veins (MPV) suggesting a variable incompletely penetrant phenotype[66], and a recent report describing a family member with PAH without ACD and two siblings with ACD[67].

Delivery of *FOXF1*, a pulmonary EC-specific transcription factor, using an AAV specifically targeting the pulmonary vascular endothelium could be a novel and attractive treatment and we could also screen for drugs that could be repurposed or small molecules that selectively increase FOXF1. AAV-based gene therapy was recently approved by the US Food and Drug Administration for spinal muscular atrophy and retinal dystrophy and presents a promising avenue for the treatment of PAH. Other agents, including nanoparticles to activate STAT3 downstream of FOXF1, have shown feasibility in experimental studies by the Kalinichenko group[68].

## Methods

### Human pulmonary artery endothelial cells (PAEC)

Primary human PAEC were purchased from PromoCell (#C-12241) and grown in complete EC media (EC basal medium #1001, ECGS #1052, 5% FBS #0025, and Penicillin/Streptmycin #0503) (ScienCell). The medium was changed every 48 h. Cells (passage 4–7) were used to experiment. For hypoxia-reoxygenation studies, human PAEC were incubated under hypoxia (0.5% $O_2$, 5% $CO_2$) in a hypoxic workstation (Invivo2 400, Ruskinn Technology Microaerobic System, Bridgend, UK) for 48 h followed by re-incubation in normoxia (21% $O_2$ 5% $CO_2$) for 24 h. For hypoxia studies, cells were harvested in the hypoxic workstation. All experiments were performed at sea level. We used 0.5% oxygen in the hypoxia chamber for in vitro experiments. The mean $PO_2$ level in the culture medium in the hypoxia chamber was 36.0-37.6 mmHg, compared to 145.7-150.4 mmHg in room-air. Given that the value generally used as the normal $PvO_2$ (mixed venous oxygen tension) level is 40 mmHg[69], our findings represent a model of relative hypoxia in EC cell cultures.

Human PAEC were also isolated from pulmonary arteries harvested from explanted lungs of PAH patients undergoing lung transplantation, all obtained deidentified from the Pulmonary Hypertension Breakthrough Initiative (PHBI) Network, supported by NIH R24 HL123767 and the Cardiovascular Medical Research and Education Fund (CMREF).

### Human lung sections

Human lung tissues were obtained from PAH and unused donor-explanted lungs as controls. The patients' characteristics and hemodynamic data are shown in Supplementary Table 3. The lungs were fixed with 10% v/v neutral buffered formalin (NBF), embedded in paraffin, and sectioned at 4-μm thickness. All samples were routinely stained with hematoxylin–eosin and Movat pentachrome stains.

### Transgenic mice

The following mouse strains were used: *Rosa*[tdTomato] Cre reporter (The Jackson Laboratory, B6.Cg-Gt(ROSA)26Sor[tm14(CAG-tdTomato)Hze]/J, Stock # 007914), *Cdh5CreER* (kindly obtained from Dr. Kristy Red-Horse[70]), *Atm*[fl/fl] (The Jackson Laboratory, 129-Atm[tm2.1Fwa]/J, Stock # 021444), *Bmpr2*[fl/fl] mice (described in our lab[21]). *Cdh5CreER* mice were bred with *Rosa*[tdTomato] Cre reporter mice. *Atm*[fl/fl] or *Bmpr2*[fl/fl] mice were crossed with *Cdh5CreER*/*Rosa*[tdTomato] mice. The number of mice for each condition tested is indicated in the figure legends. There was no attrition in our experiments and all mice were included in the data analysis. However, based on the vagaries of breeding some cohorts of mutant and wild-type mice were larger than others and our focus was largely on the phenotype in reoxygenation.

### Husbandry

The Animal Care and Use Committee at Stanford University approved all the experimental protocols (Stanford APLAC protocol 10704) used in this study in accordance with the guidelines of the American Physiological Society. Mice were housed under standard conditions according to protocols of the Stanford University Institutional Animal Care and Use Committees in pathogen-free, individually ventilated microisolator cages in a room with a 12 h light/dark cycle, and a temperature- and humidity-controlled environment of 20–26 °C and 30–70% humidity, with access to standard laboratory chow diet and water ad libitum.

### Hypoxia-reoxygenation mouse model

For Cre induction, tamoxifen (Sigma-Aldrich T5648) was dissolved in corn oil (Sigma-Aldrich #C8267) at a concentration of 20 mg/ml and

was injected into the peritoneal cavities at 2 mg per day for 8 days; after another 7 days, Cre activation was assessed and deletion in *Bmpr2* or *Atm* in EC in lungs was observed. *Cdh5CreER/ Rosa^tdTomato* mice were injected with tamoxifen and used as controls. The mice ranged in age from 7-10 weeks at the start of the experiment and were placed in a hypoxia chamber. $O_2$ level was adjusted to 10% by an $O_2$ controller (ProOx 360; BioSpherix) and a sensor (Telaire; Amphenol) for three weeks, followed by four weeks of recovery in room air. All experiments were performed at sea level. The $O_2$ level and the condition of the mice were monitored daily and represent an altitude of about 19,000 feet that is well tolerated by mice.

### Hemodynamic measurements

Cardiac output and left ventricular function (fractional shortening) as well as pulmonary artery acceleration time (PAAT) were evaluated by echocardiography under isoflurane anesthesia (1.5–2.5% in 1 L $O_2$/min) using a Vivid 7 ultrasound machine (GE Healthcare) or Vivo2100 (FUJIFILM VisualSonics). The left ventricle (LV), diastolic and systolic diameters, heart rate (HR), the velocity time integral (VTI) of the LV outflow tract, the pulmonary arterial acceleration time (PAAT) and ejection time (ET) were measured. The percentage of the left ventricular ejection fraction (%LVEF) was calculated as described by Stypmann et al.[71]. Left ventricular cardiac output (LVCO) was calculated as: [(aortic diameter/2)$^2$ x 3.1416] x VTI x HR. Mice were monitored during the echocardiographic assessment, as LVCO is distinctly dependent on heart rate. The following day, right ventricular systolic pressure (RVSP) was measured in unventilated mice under isoflurane anesthesia (1.5–2.5% in 1 L $O_2$/min). 1.4-F Millar catheter (Millar Instruments, Model SPR-671) was inserted into the jugular vein and directed to the right ventricle. Pressure measurements were repeated three times and averaged. Data were collected by Power Lab Data Acquisition system (AD Instruments) and analyzed using LabChart software (AD instruments). The mice were sacrificed by cervical dislocation under anesthesia and perfused with cold PBS and then the heart and lungs were harvested. The left lungs were fixed with 10% v/v neutral buffered formalin for 24 h, and then embedded in paraffin for immunohistochemistry. Right ventricular hypertrophy (RVH) was evaluated as the weight of the right ventricle (RV) relative to the weight of the left ventricle plus septum (LV + S).

### AAV administration to mice

The pulmonary EC targeting AAV2-ESGHGYF vector from the Körbelin Lab (University Medical Center Hamburg-Eppendorf, Germany[23]) was used for all AAV experiments. Recombinant AAV2-ESGHGYF particles carrying mouse *Foxf1* [NM_010426.2] or luciferase as a control were obtained from VectorBuilder. The *Foxf1* or luciferase AAV vectors were injected into the tail vein at a dose of $5 \times 10^{10}$ vector genomes (vg) in 100 µl PBS per mouse. On days 7 and 28 after administration, animals injected with luciferase control vector were anesthetized with isoflurane (1% in 1 L $O_2$/min) and Luciferin substrate (Biosynth L-8220) was injected into the intraperitoneal cavity of mice at 150 mg/kg, and bioluminescence was visualized 20 min after injection with the in vivo imaging tool (LagoX; Spectral instruments imaging).

### Blood collection

The mice were anesthetized, and blood was drawn from the superior vena cava with 24 G needles and then sacrificed. The blood samples were analyzed by the Animal Diagnostic Lab at Stanford University.

### Mouse lung EC isolation

Right lungs pooled from 2–3 mice were rinsed with ice-cold PBS and transferred into a gentleMACS C tube (Miltenyi Biotec #130-096-334) containing 6 ml medium (DMEM (Thermo Fisher Scientific #10829018) supplemented with 6 mg collagenase II (Gibco #7101015). Each sample was dissociated using the gentle MACS Dissociator system (MACS

Technology, Miltenyi Biotec). First, samples were run using the m_lung_01 protocol (pre-programmed by the manufacturer) and then the samples were placed in a rotator at 37 °C for 30 min. After the incubation, samples were again processed using the MACS Dissociator system using the m_lung_02 protocol (pre-programmed by the manufacturer). The cell suspension was placed on ice, neutralized with cold PBS with 5% BSA, and filtered through a 70 µm cell strainer (Corning # 352350). The cell suspension was then centrifuged at 300 x g for 5 min, and then washed with cold PBS. Next, the cell suspension was depleted of CD45-positive cells using CD45 magnetic beads (MACS Technology, Miltenyi Biotec #130-052-301) according to the manufacturer's instructions. The CD45 depleted single-cell suspension was enriched for EC using CD31 MicroBeads (Miltenyi Biotec #130-097-418) according to the manufacturer's instructions. CD45⁻ CD31⁺ ECs were washed with PBS and RNA was extracted.

### Reverse-transcriptase qPCR (RT-qPCR)

Total RNA was extracted and purified from cells using the Quick-RNA MiniPrep Kit (Zymo Research #R1055). The quantity and quality of RNA was determined using a Synergy H1 Hybrid Reader (BioTek) and then RNA was reverse transcribed using the High Capacity RNA to cDNA Kit (Applied Biosystems #4387406) according to the manufacturer's instructions. RT-qPCR was performed using 1 µl of 5 µM mixed primers, 5 µl Power SYBR green PCR Master Mix (Applied Biosystems # 4309155), 2 µl dH$_2$O and 2 µl cDNA sample. The qPCR program consisted of 50 °C for 2 min, 95 °C for 30 sec, followed by 40 cycles of 95 °C for 15 sec, and 60 °C for 60 sec. Each measurement was carried out in duplicate or triplicate using a CFX384 Real-Time System (Bio-Rad) in a 10 µl reaction and performed according to the manufacturer's instructions.

Primer sequences were designed using NCBI's Primer-BLAST function or PrimerBank (http://pga.mgh.harvard.edu) and are as listed in the table below. Data were analyzed by the delta-delta CT method. Expression levels of selected genes were normalized to β-Actin or GAPDH. Primer sequences used are listed In Supplementary Table 4.

### Bulk RNA-seq sample preparation and data analysis

The RNA isolate was thereafter enriched for poly-A templates and libraries were prepared with CORALL mRNA-seq library prep kits (Lexogen, #162-96) and subjected to sequencing with an Illumina NovaSeq 6000 instrument (Stanford Genomics, supported by NIH grant S10OD020141). Sequencing on the Illumina NovaSeq 6000 yielded an average of approximately 25 million uniquely mapped reads per sample for RNA-Seq for each sample. The FASTQ reads were QC-checked using FastQC v0.11.9 and underwent quality and adapter trimming using Trim Galore v0.6.6. The resulting FASTQ reads were analyzed using the nf-core/rnaseq pipeline v3.3[72] (https://nf-co.re/rnaseq) and aligned to the mouse reference genome mm10 using STAR v2.6.1d. The aligned transcripts were quantitated by RSEM v1.3.1 (http://deweylab.github.io/RSEM/). Differentially expressed genes were detected using DESeq2 v 1.32.0 (https://bioconductor.org/packages/release/bioc/html/DESeq2.html) with an adjusted p-value cutoff of <0.05. GO enrichment analysis was performed for 512 commonly upregulated genes and 547 commonly downregulated genes using clusterProfiler v4.4.1 (https://bioconductor.org/packages/release/bioc/html/clusterProfiler.html) with a q-value cutoff of <0.05.

Murine *Foxf1* target genes were downloaded from ChIP-Atlas database (https://chip-atlas.org)[39] of the mouse reference set (mm10). We found 11,222 genes with *Foxf1* peaks at a distance of +/−1 kb or less from the transcription start site (TSS). We also used a gene list associated with DNA repair in mice (GO 0006281) showing 577 genes.

### Single-cell RNA-seq analysis

**Sample preparation.** The right lungs of 3 EC-*Bmpr2*$^{/-}$ mice or 4 control mice following reoxygenation were pooled for single-cell RNA-seq

analysis. Isolated right lungs were transferred into a gentleMACS C tube containing digestion medium 0.4 mg/ml Liberase (Sigma #5401020001) and 10 mg/ml elastase (Worthington #LS002292) in 6 ml of RPMI-1640 medium. Each sample was dissociated using the gentle MACS Dissociator system following the manufacturer's protocol. Each sample was run using the m_lung_01 protocol and placed in a rotator at 37 °C for 30 min followed by dissociation with the MACS Dissociator system using the m_lung_02 protocol. The cell suspension was placed on ice, neutralized with cold PBS with 5% BSA, and filtered through a 70 μm cell strainer. The cell suspension was then centrifuged at 300 x $g$ for 5 min, and then washed with cold PBS. Next, the cell suspension was depleted of CD45-positive cells and enriched for EC using CD31 microbeads according to the manufacturer's instructions and collected in cold PBS with 2% BSA. Cell viability was determined using trypan blue staining and cells with viability >90% were sent to Stanford Genome Sequencing Service Center for 10X chromium single-cell RNA-seq paired end library preparation (Chromium Single Cell 3' Reagent Kits v3.1; 10X Genomics). Approximately 10,000 cells were loaded for each sample, targeting cell recovery of 6000 cells. The libraries were sequenced with NovaSeq 6000. The sequencing depth of each sample was around 30,000 reads per cell.

**Single-cell RNA-seq analysis.** Single-cell RNA-seq outputs were processed using the CellRanger (10X Genomics, version 6.0.0) followed by mapping with mm10 reference genome and added sequences of tdTomato, obtained from Dr. Hongkui Zeng (Allen Institute for Brain Science) who originally created the $Rosa^{tdTomato}$ Cre reporter mice (Ai14) obtained from the Jackson Laboratory. We annotated these sequences in the GTF (gene transfer format) file and rebuilt the reference genome by the mkref command implemented in the CellRanger software. To retain high quality single cells, we excluded cells with greater than 50,000 UMIs per cell, and cells expressing less than 500 or greater than 8000 genes. We also excluded cells that had more than 10% mitochondrial genes per cell or more than 50% ribosomal genes per cell, to avoid analyzing doublets or low-quality cells. We retained cells expressing tdTomato ( > 0). We found a trivial number of tdTomato-negative cells ($n$ = 219 vs. 12,469 tdTomato-positive cells) in EC-$Bmpr2^{-/-}$ mice that expressed low levels of the endothelial marker $Cldn5$. The remaining matrix was log-normalized with a scale factor of $10^4$. Expression profiles of cells were clustered using the R software package Seurat (v.3.0.2). Highly variable genes were selected using the 'FindVariableGenes' function for linear dimensionality reduction using principal component analysis. The number of principal components was selected by inspection of the plot of variance explained. Principal components were examined using functions 'ElbowPlot' and 'DimHeatmap' to determine the number of principal components for downstream analysis. Cells were clustered by constructing a shared nearest neighbor graph performed using the 'FindNeighbors' function and clusters were visualized by Uniform Manifold Approximation and Projection (UMAP). The clusters of murine lung EC subpopulations were assigned using expression of cell type markers for artery ($Bmx$), vein ($Slc6a2$), aerocytes ($Ednrb$), general capillary ($Aplnr$), and lymphatic endothelial cells ($Prox1$) previously shown by Gillich et al.[41] and confirmed in public available data shown (https://tabula-muris.ds. czbiohub.org)[43] in Supplementary Fig 4. Differential gene expression analysis comparing Control reoxygenation and EC-$Bmpr2^{-/-}$ reoxygenation mice was performed by Wilcoxon rank sum test within each cell subpopulation and reported $P$-values.

**Flow cytometry analysis.** Control mice ($Cdh5CreER/Rosa^{tdTomato}$) injected with tamoxifen were treated with AAV-luciferase after 3 weeks of hypoxia (10% $O_2$). Lungs were harvested one week after AAV injection and digested enzymatically as described for mouse lung EC isolation. A single-cell suspension was prepared by passing the digested tissue through a 70 μm nylon strainer. Red blood cells were lysed in

RBC lysis buffer (eBioscience # 00-4333-57), the suspension was washed with cold PBS, and centrifuged at 300 x $g$ for 5 min. Single cell suspensions were incubated with TruStain FcX (Biolegend #101319) and stained with fixable viability dye (Biolegend # 423101) followed by fixation and permeabilization (BD # 554714). The cells were stained with luciferase antibody (1:200 Abcam #Ab185923) at room temperature for 30 min followed by staining of Alexa fluor 647 (1:1000). Data were acquired using BD FACS Aria cytometer and analyzed using FlowJo software version 10.7.1. Gating strategies are provided in the Supplemental figures. Single cells determined by forward and side scatter gating strategy were analyzed, and dead cells were excluded from analysis. In flow cytometry, lung EC were identified by tdTomato fluorescence.

**siRNA transfection.** Before transfection, human PAEC were incubated with Opti-MEM (Life Technologies) for 30 min. The siRNA for human $BMPR2$ (ON-TARGETplus SMARTpool #L005309-00-0050, Dharmacon), human $ATM$ (Silencer select #s530445, Life technologies) and human $FOXF1$ (Silencer select #s5220, Life technologies)) were transfected into human PAEC at the concentration of 10 nM by using lipofectamine RNAimax (LifeTechnologies #13778150) according to manufacturer's recommendations. Cells were incubated for 6 h at 37 °C, and then returned to ECM medium. After 48 h, the knockdown efficiency was determined by immunoblotting or qPCR. Non-targeting control siRNA (Dharmacon, ON-TARGETplus Non-targeting Control Pool #D-001810-10) was used as siControl. The level of knockdown was similar at days 2, 4, and 6 for each target.

**Alkaline comet assay.** DNA damage in individual cells was assessed using the Comet Assay Reagent Kit for Single Cell Gel Electrophoresis Assay (Trevigen #4250-050-K) according to the manufacturer's protocol. Human PAEC were collected and suspended in cold PBS at $3 \times 10^5$ cells/ml. Cell suspension (50 μl) was mixed with 500 μl LMA Comet agarose (1:10, v/v) in an Eppendorf tube and inverted 5 times to mix. 50 μl of mixture was immediately placed on the Comet kit slide. Cells were embedded on the agarose by cooling the gel at 4 °C for 1 h in the dark and then lysed in pre-chilled lysis solution at 4 °C overnight. The cell-containing slides were immersed in alkaline unwinding solution for 1 h at room temperature and then carefully transferred to a horizontal electrophoresis chamber. Electrophoresis was conducted in cold Alkaline Electrophoresis Solution for 30 min at 300 mA/30 V current. The slides were washed two times in water for 5 min and immersed in 70% ethanol for 5 min. The slides were dried on a heat block at 37 °C. DNA was stained with 100 μl/well of diluted SYBR-gold (Invitrogen #S11494) (1:10000 TrisEDTA, pH 8). After incubating at room temperature in the dark for 5 min, slides were visualized using a Leica DMLB Fluorescence Microscope (Leica). The comet tail lengths (defined as the length from the center of the DNA head to the end of the DNA tail) of each cell were quantified using the OpenComet software[73] (ImageJ (v. 2.0) with a Comet Assay Plugin, downloaded from https://www.med.unc.edu/microscopy/resources/imagej-plugins-and-macros/comet-assay). For each sample, 100–200 cells were analyzed and the box in box-and-whiskers plots corresponds to the 25th to 75th percentiles. The line in the box marks the median and whiskers correspond to the 10th to 90th percentiles. The outliers are represented as dots outside the whiskers.

**Tube formation assay.** Matrigel (10 μl; Corning # 356231) thawed at 4 °C was applied to each inner well of a slide (Ibidi, μ-Slide Angiogenesis # 81506) placed on ice. The slide was placed in the petri dish with water-soaked paper towels to maintain humidity and incubated for 30 minutes in a 37 °C incubator. Human PAEC were collected, diluted with 5% FBS ECM medium to $2 \times 10^5$ cells/ml, and 10,000 cells (50 μl) were seeded per well and incubated at 37 °C for 6 h. The number of tubes were assessed by a microscope (ECHO Revolve) and counted

using ImageJ. The bright field images were inverted using ImageJ for clarity of the figures presented.

**Migration assay.** The migration of human PAEC was assessed using a scratch wound assay[74]. Forty-eight hours after siRNA transfection, a linear scratch ("wound") was generated in the monolayer with a sterile p200 pipette tip. Cellular debris was removed by washing with PBS and then 5%FBS ECM media were added. After incubation for 12 h, five representative images of the scratched areas from each well were photographed to calculate the wound area using ImageJ. For clarity of the figures presented, the bright field images were inverted using ImageJ.

**Cellular ROS measurement.** After siRNA transfection, PAEC were cultured in a 6-cm dish in the hypoxia chamber (0.5% oxygen) for 48 hours. For the assay, hypoxia samples were incubated in a normal ECM medium containing 2.5 μM CM-H2DCFDA (C6827; Thermo-Fisher) for 15 min at 37 °C. For samples of reoxygenation (15 min, 30 min and 60 min), the cells were transferred to normoxia and incubated in a normal ECM medium containing 2.5 μM CM-H2DCFDA. The cells were washed twice with PBS and then trypsinized. Fluorescence was measured by flow cytometry at the Stanford FACS facility using a cytometer FACS Aria II instrument (BD Bioscience) and analyzed using FlowJo version 10.7.1 software for mean fluorescent intensity (MFI).

**Western immunoblot.** The cultured cells were washed with ice-cold TBS, and lysates were prepared by adding lysis buffer (50 mM Tris-HCl (pH 8.0), 150 mM NaCl, 5 mM $CaCl_2$, 1% NP-40, 0.5% sodium deoxycholate, 0.1% SDS, protease and phosphatase inhibitor) incubated with micrococcal nuclease (Thermo Scientific # 88216) at room temperature for 15 min. Cell extracts were collected by centrifugation at 20,000 x g at 4 °C for 15 min. Protein concentration was determined by the BCA assay (Thermo Scientific #23228). Equal amounts of protein were mixed with sample buffer (Thermo Fisher Scientific NuPAGE LDS Sample Buffer #NP0007) containing TCEP (Pierce #77720) and were separated by SDS-PAGE on Bolt 4%–12% Bis-Tris Plus gels (Thermo Scientific # NW04122) and transferred onto polyvinylidene difluoride (PVDF) membranes (Biorad #1620177) at 120 V for 90 min at 4 °C. PVDF membranes with proteins were washed with 0.1% Tween-20 in Tris-buffered saline (TBST) for 10 min and blocked with TBST + 5% bovine serum albumin (BSA; sigma Aldrich #A3059) at room temperature for 1 h. Next, the membrane was incubated in 5% BSA-TBST with primary antibodies overnight at 4 °C. After overnight incubation, the membrane was washed three times with TBST for 15 min and incubated with secondary horseradish peroxidase (HRP) antibody (1:5,000) for 1 h at room temperature. Membranes were developed using chemiluminescence reagent Clarity Western ECL Substrate (BioRad #1705061) or SuperSignal West Femto Maximum Sensitivity Substrate (Thermo Scientific #34096) with BioRad ChemiDoc XRS system. Densitometric quantifications were performed using ImageJ. For data using cell lines, quantification is presented relative to the values of control normoxia samples or relative to the minimum value in the control samples in Supplementary Figs. 1b and 2b. For mouse lung EC in Fig. 7g, quantification is presented relative to the value of a normoxia control pooled sample assessed in triplicate. Antibodies for western immunoblotting are listed in Supplementary Table 5. To avoid the need for stripping and reblotting, the full membrane was divided into different sections by molecular weight and the sections were then incubated with specific antibodies according to the molecular weight of the proteins being assessed. The uncropped and unprocessed images of those gel sections are provided next to the gel as it appears in the manuscript and supplement, in the Source Data file.

## Immunostaining

**Immunohistochemistry (IHC).** Mouse sections from paraffin-embedded lung tissues were de-paraffinized and rehydrated. Epitope retrieval was performed by boiling the sections in citrate buffer (pH 6.0, sigma Aldrich #C999) for 20 min. Sections were incubated with 0.3% hydrogen peroxide in methanol for 30 min to block endogenous peroxidase, washed, and blocked with 5% BSA + 0.2% Triton X-100 in TBS (TBST) for 1 h at room temperature. The sections were incubated with primary antibody αSMA (Sigma # A2547) by using Dako animal research kit (Dako #A3954) under the manufacturer's instructions. The stained sections were developed with DAB+ substrate-chromogen for 2 min and counterstained with hematoxylin. Images were acquired using a BZ-X710 microscope (Keyence) with 20x objectives. Muscularization was determined by αSMA staining of the vessels and assessed by comparing the number of muscularized vessels to the total number of distal small vessels at alveolar duct and wall level (diameter <50 μm). The number of these small pulmonary vessels per 100 alveoli was assessed by comparing the number of vessels to the number of alveoli in same image.

**Immunofluorescence staining (IF).** Lung sections were de-paraffinized, and epitope retrieval was performed by boiling for 20 min in citrate buffer for luciferase staining or in 0.25 mM EDTA pH 8 (Invitrogen #15575-038) for γH2AX and FOXF1 staining. The sections were washed with TBS and blocked with 5% BSA-TBST for 1 h at room temperature, and then incubated with the primary antibodies (details below) in 1.5% BSA-TBS overnight at 4 °C. The next day, the sections were washed three times with TBS for 5 min and then incubated with the secondary antibody (1:500 in 2% BSA-TBST) for 1 h at room temperature. The sections were washed three times with TBS and then mounted with Vectashield containing DAPI (Vector Laboratories #H1200). Images were obtained using a confocal microscope (Zeiss, LSM880) and analyzed with ZEN software.

To count γH2AX foci in mouse lung sections, 5 vessels (approximately 50 μm in diameter) per sample were analyzed in 6 mice per group. To count γH2AX foci in αSMA-positive SMC in the control normoxia group, we examined proximal vessels with diameters of 100-200 μm, since mice in this group had little distal arterial muscularization. γH2AX foci were counted in three kidney arteries, and in five coronary arteries per mouse. In the case of the aorta, foci were determined in five different positions per mouse (6 mice per group). Images were acquired with X40 objectives. A number of nuclear γH2AX foci in each tdTomato-positive EC was measured using the Foci-Counter software[75] and normalized to cell number per vessel. Each data point represents the average nuclear γH2AX foci (after normalization to cell number) per vessel.

For FOXF1 intensity for mouse and human lung sections, 5 vessels per sample were analyzed (6 mice or 6 subjects per group). Images were acquired with 40X objectives. In the murine lung sections, nuclear FOXF1 intensity of tdTomato positive cells in a vessel was analyzed using ImageJ[76], and normalized to cell number per vessel. Each data point represents the average nuclear FOXF1 intensity per vessel. For human lung sections, nuclear FOXF1 intensity of vWF-positive cells in a vessel was similarly analyzed.

**Immunocytochemistry (ICC).** A day before fixation, cells were seeded at $5 \times 10^3$ per well in 8-well chamber slides (Thermo Fisher Scientific, LAB-TEK II chamber slide #154534) and incubated overnight. The next day, cells were fixed with 4% paraformaldehyde for 10 min, and blocked in 1.5% BSA-TBST for 1 h. Cells were stained with γH2AX and pRPA primary antibodies (1:300 in 1.5% BSA-TBS) overnight at 4 °C and Alexa Fluor 488 or 568 secondary antibodies (Invitrogen) for 1 h at room temperature, washed, then mounted with DAPI with Vectashield containing DAPI. One to 3 fields were acquired with the 20X objective to obtain 100–150 cells per sample. Quantification of the nuclear

fluorescence intensities was measured using ImageJ[76]. Antibodies used for immunostaining are listed in Supplementary Table 6.

**Cell transfection.** Lentiviral vector *FOXF1* (pLV-CMV-human-FOXF1[NM_001451.3]-EGFP-puro) (Vector ID: VB211013-1245zbb) or empty *GFP* control vector were obtained from VectorBuilder. The pLV vectors were co-transfected with packaging plasmids (Celecta #CPCP-K2A) into HEK 293 T cells using Lipofectamine 3000 reagent (Invitrogen # L3000001) according to the recommendations of the manufacturer. Twenty-four hours after transfection, media was changed to 5% FBS ECM media and the viral supernatant was harvested 24 and 48 h later. For infection, virus supernatants were added to human PAEC together with fresh ECM media with 8 μg/mL polybrene (Santa cruz #sc-134220), and fresh ECM media were added the next day. Infected cells were selected with puromycin (1 μg/ml). Transduction efficiency was confirmed by immunoblotting.

**Statistical analysis.** All data are expressed as mean ± standard error of the mean (S.E.M.). Statistical significance was determined by Two-sided unpaired t-test analysis used for the comparison of two groups, by one-way ANOVA with Holm-Sidak posthoc test for comparison of three groups and by 2-way ANOVA with Holm-Sidak posthoc test for four or six groups with two variables. For comparisons of immunofluorescence signals or the comet assay, Kruskal–Wallis test followed by Dunn's test was used. Analyzes were carried out using GraphPad Prism 9.0 (GraphPad Software Inc.). A *P*-value of <0.05 was considered significant. The number of experiments and the number of animals in each group are indicated in the figure legends. *P*-values for comparisons are indicated in the figures. For analyses by t-test and ANOVA, the GraphPad Prism program provides exact *P* values for $P = 0.0001$ or higher; for lower values, it yields $P < 0.0001$.

**Human samples.** Where indicated, the donor of primary human pulmonary artery endothelial cells (PAEC) commercially obtained from PromoCell (C-12241) was male. As indicated in the text we also used lung tissues and PAEC from healthy donors (five males and four females), and from PAH patients (four males and ten females). Sex was provided by the procuring center.

**Study approvals**
**Animal studies.** The Animal Care and Use Committee of Stanford University approved all protocols involving mice, in keeping with the regulations of the American Physiological Society.

**Studies with human cells and tissues.** Cells purchased from Promo-Cell were derived from the tissues of donors who have signed an informed consent form. Human tissues provided by the PHBI Initiative were obtained under the PHBI Network protocol, approved by the Medical School IRB at the University of Michigan, and written informed consent at the lung procurement sites. There was no compensation to the donors for participation in the study. The cell lines used were coded with no identifying information. The informed consent outlined in detail the purpose of the donation and the procedure for processing the tissue.

**Reporting summary**
Further information on research design is available in the Nature Portfolio Reporting Summary linked to this article.

## Data availability

Bulk RNA-Seq and single-cell RNA-seq data generated by this study have been deposited in the Gene Expression Omnibus (GEO), under accession code GSE215933. Potential *Foxf1* target genes were provided by ChIP-Atlas database (https://chip-atlas.org/target_genes). Mouse lung scRNA seq data in Supplementary Fig. 5 was provided by Tabula Muris (https://tabula-muris.ds.czbiohub.org). Source data are provided with this paper.

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

## Acknowledgements

We thank our Stanford colleagues: Dr. Christin Kuo for the use of the Zeiss confocal microscope and Dr. Amato Giaccia for the use of hypoxia chamber; Patricia A. del Rosario for retrieving clinical information of the PHBI samples; Michelle Ameri for administrative assistance; Dr. Michal Bental Roof for editorial help with the manuscript and the figures. Sequencing was performed by the Genomics and Personalized Medicine Sequencing Center, supported by award number NIH S10OD020141. This work utilized computing resources and computational and bioinformatics services provided by the Stanford Genetics Bioinformatics Service Center (GBSC). Flow cytometry analysis was performed at the Stanford Shared FACS Facility. The Pulmonary Hypertension Breakthrough Initiative (PHBI) Network was funded by NIH/National Heath, Lung, and Blood Institute (NHLBI) R24 HL123767 and the Cardiovascular Medical Research and Education Fund (CMREF) grant UL1RR024986. De-identified demographic and clinical data of PHBI samples were provided by the Data Coordinating Center at the University of Michigan. Some images presented in Figs. (1d, 2d, 4a, 5a, 7a, 8) and Supplementary Figs. (1a, 1g, 2g, 7a, 7b) were created with BioRender.com. This work was supported by NIH/NHLBI grant R01 HL087118 (MR); The MSD Life Science Foundation Fellowship Grant, American Heart Association Postdoctoral Fellowship Award 20POST35080009 and Japan Society for the Promotion of Science Postdoctoral Fellowship for Research Abroad (SI); The California TRDRP award 27FT-0039 and the Netherlands Heart Foundation award 2013T116 (JRM); NIH/NHLBI T32 HL129970 fellowship and NIH/NHLBI Research Diversity Supplement P01 HL108797-04W1 (ST); NIH-NHLBI grant K99 HL1450970 (DPM); a Postdoctoral Fellowship from the Fundación Ramón Areces (MLD). JME acknowledges support from an NIH Pathway to Independence Award (K99 HG009917 and R00 HG009917); NHLBI R01 HL152134; Gordon and Betty Moore; and the BASE Research Initiative at the Lucile Packard Children's Hospital at Stanford University. Dr. Marlene Rabinovitch is additionally supported by the Dwight and Vera Dunlevie Chair in Pediatric Cardiology at Stanford University.

## Author contributions

S.I. and M.R. conceived the project, designed the research studies, and prepared the manuscript. S.I., L.W., T.S. and S.O. performed the experiments and analyzed the data. R.V.N. analyzed bulk RNA sequencing data. R.V.N. and H.Y.K. analyzed single-cell RNA sequencing data. J.R.M., A.C., S.T., D.P.M. and R.L.H. provided technical assistance. M.A., C.Z. and M.L.D. assisted with qPCR and western immunoblotting. J.K. provided the AAV peptide. M.P.S. and J.M.E. assisted in the supervision of the project.

## Competing interests

Jacob Korbelin is an inventor of, and received royalties for, a patent on the capsid-modified AAV-ESGHGYF vector, granted to Boehringer Ingelheim International GmbH. Jesse M. Engreitz is a consultant and equity holder in Martingale Labs, Inc. and has received materials from 10x Genomics unrelated to this study. The remaining authors declare no competing interests.
