## [Peer Review File · Nature Communications]

REVIEWER COMMENTS

Reviewer #1 (Remarks to the Author):

Dear colleagues,

I have read with interest your paper on role of ATM BMRP2 and Foxf1 in endothelial cells DNA damage signalling in pulmonary hypertension.

I found the paper very well written logical and easy to follow. I have several minors comments.

1) given the model used hypoxia followed by reoxygenation I suppose that authors believe that DNA damage results from oxidative damage due to ROS ? Could authors monitor ROS levels during the course of the experiments ?

2) given that multiple DNA damage pathways are affected in PHT it will be interesting to assess the expression levels of couple of them including DNA PK which as just been showed to be important in PAH (January 2023 paper in the red ATS journal)

3) can authors speculate what drive FOXF1 down regulation? In PAH tissues is it specific to EC?

4) given that Foxf1 regulates fanconi complex are the members of the complex affected in EC

5) why using hypoxic mice followed with reoxygenation and not sugen ? This experimental model seems to be closer to group 3 PHT than group 1 according to Boucherat review in Circulation research on animal model.

6) is there any alteration in DNA damage response in ECs isolated from systemic vessels or is it specific to the lungs because of hypoxia and reoxygenation? Any systemic abnormalities

7) although mice models are EC specific any alteration in PASMC or fibroblasts? Just wondering here if the presence of damage ECs can affect the functioning of the surrounding PASMCs.

8) as much as possible try to reach sample size of greater than 5,

9) comet assay should be presented by the mean of tail length by patients or mice not the total cell count which will artificially increase power.

Reviewer #2 (Remarks to the Author):

In this manuscript, Isobe et al investigated a causal relationship between DNA damage and the development of pulmonary arterial hypertension (PAH). The premise stems from a large set of previously published work that links BMPR2 to DNA damage repair. In the current study, the authors found that FOXF1 transcription factor is decreased in endothelial cells from 2 different animal models that use the exposure to hypoxia-reoxygenation cycle to cause PAH. Importantly, the authors demonstrated that FOXF1 gene replacement therapy was sufficient to ameliorate the pulmonary hypertensive phenotype in both animal models. I commend the authors for the amount of work performed in this project and congratulate them for generating such a comprehensive study with significant clinical implications. The manuscript contains high-quality experimental data, and main conclusions are supported by the data. However, the manuscript can be improved by addressing the following issues:

Major points:

1. Do ATM^{-/-} mice exhibit similar vascular remodeling compared to BMPR2 mice? Do they develop vessel obliteration or any other type of complex vascular lesions?
2. Is there reduced FOXF1 expression at baseline in both models? If not at baseline, can the authors speculate or provide the data to demonstrate at which time point FOXF1 downregulation occurs?
3. Please provide a scatter plot correlating RVSP with RV/LV+S. Showing group data is useful for intergroup comparisons however it does not allow to see the linear relationship that is expected from the experiment.
4. Is there a difference in FOXF1 expression between TdT positive versus negative cells? Having single cell sequencing data provides an opportunity to truly evaluate the transcriptional profile differences between lacking BMPR2 (positive Cre recombination, TdT positive) and cells with presumably normal BMPR2 (TdT negative).
5. Flow cytometry data are needed to show specific targeting of endothelial cells and off-target cell selection. AAV with GFP in wild type s/p hypoxia mice should be sufficient. IF alone is insufficient evidence.
6. It is possible that the phenotype after AAV-Foxf1 treatment is secondary to new vessel formation (thus dropping pulmonary vascular resistance) rather than reversal of established pathological vascular remodeling. This possibility should be addressed in the Discussion section.

7. It will be helpful to provide additional data on RV function (either TAPSE or CO) for treatment groups.

Minor points:

1. Page 6 line 124: It is unclear if Cdh5-CreER/Rosa-TdTom mice were treated with tamoxifen and used as controls or the mice without tamoxifen treatment were used as control. Please specify.

2. There is a mistake (Line 80, page 4, introduction): "...persistent DNA damage in ECs and pulmonary arterial hypertension results from.."

3. Page 6, line 121: The reference for the mouse model cited goes to a paper where the mouse model is cited to a different paper by Spiekerkoetter. The mouse model by Spiekerkoetter in JCI paper is a SCL-CreERTM+/R26R/Bmpr2-/- . The authors state they are using a Cdh5-CreER/Rosa-TdTom. These are effectively 2 different models. Please, correct this error?

4. I commend the authors for generating a visually appealing graphical abstract. However, some information is misleading. As portrayed, it appears as if PAH secondary to BMPR2 dysfunction results from capillary abnormalities and perhaps arterialization of veins or even A-V shunts which perhaps is more common in ACDMPV rather than PAH.

Reviewer #3 (Remarks to the Author):

General comments:

In this study, Isobe et al. investigated the cellular mechanisms underlying persistent pulmonary hypertension (chronic hypoxia-induced PH that persisting after reoxygenation) in endothelial cells (EC) specific-Bmpr2-/- mice. In those mice, reoxygenation after chronic hypoxia, in contrast to control mice, did not reverse pulmonary hypertension, which was associated with increased DNA damage. The deletion of Atm in EC (EC-Atm-/- mice) mimicked the EC-Bmpr2-/- mice phenotype after reoxygenation. Transcriptome analysis showed that downregulation of the transcriptional factor Foxf1 can be cause of persistent pulmonary hypertension in EC-Bmpr2-/- and EC-Atm-/- mice. Restoration of Foxf1 level during reoxygenation reversed persistent pulmonary hypertension. These findings are very interesting.

Major comments:

Is there a difference between Foxf1 and/or Atm expression between EC of PH patients with and without BMPR2 mutation. Your data claim that Atm and Foxf1 is linked to BMPR2 – Thus, a difference in their expression between +/-BMPR2 mutations in PH patients would be expected.

Have statistical comparisons between all groups in the in vivo experiments performed? E. g. Figure a, b: it is not clear if a difference exists between “normoxia”, “hypoxia” and “reoxygenation” within one genotype. As an example, the muscularization in figure 3a for the Atm knockout mice seems to be decreased in reoxygenation compared to hypoxia. This looks similar for all other parameters of PH. In this regard it would be interesting to know if PH indeed persists in reoxygenation or if the recovery is delayed only. What would happen, if you follow the mice upon reoxygenation for a longer duration? This is critical to judge on the interpretation if PH is indeed persistent or recovery is delayed only.

Why are there largely differing number between groups. As an example, in figure 3 you show n=4-8 – the hypoxia Atm group consist of only n=4, the respective WT has a higher number and this differs from the reoxygenation groups. Please state for all in vivo mouse experiments how many animals were allocated to which group at the beginning of the experiment and if the group size is different, why.

The authors demonstrated that hypoxia alone (without reoxygenation) induced DNA damage of endothelial cells (proven by a higher level of H2AX and comet tail length in Supplementary figure 1 and 2) - it will be interesting to investigate the level and role of Foxf1 in EC during chronic hypoxia. Does the downregulation of Foxf1 play a role not only in the repair mechanism during reoxygenation after exposure to chronic hypoxia, or also in development of chronic hypoxia-induced PH?

Please show the basal level of Foxf1 and Atm in EC isolated from patients with PAH compared with control EC.

The spatial expression of Atm in PAH is not clear. Does Atm expression change in PAH? In which cells did Atm expression change in PAH?

Why was 0.5 %O₂ used for EC culture. This does not reflect the in vivo situation of EC pO₂. Is there a difference if you use physiological relevant pO₂ levels.

Minor comments

To analyse the data from the WB (e.g. Figure 1a) it seems normalization was performed. The normalization protocol must be described in methods or in labelling of x-axis.

If you give the hypoxia level in %, please add the information (e.g. in the methods) at which altitude the experiments were performed.

Reviewer #4 (Remarks to the Author):

The manuscript by Isobe and colleagues is a very nice paper that seeks to establish a role for the FOXF1 transcription factor in the correction of Pulmonary Artery Hypertension (PAH) with a further implication that DNA damage is among the molecular pathological insults that drive disease. To this end, they have used 2 separate animal models of PAH as well as multiple cellular studies with Pulmonary Artery Endothelial cells from diseased and control sources. They describe the consistency of FOXF1 downregulation with PAH induction as well as the extent to which restoration of FOXF1 activity via transfection can rescue the dysfunctional cellular physiology in EC cells as well as reversal of the PAH pathophysiology in their animal models.

Major comments:

None

Minor comments:

There is an opportunity to further explore the gene expression data with a more directed focus on FOXF1, which may add weight to the proposition of FOXF1's role. Both Seq and single cell datasets are often rich beyond our ability to make meaningful interpretations, and the authors provide substantiation for the role of FOXF1 from a handful of candidate target genes. However, the interrogation of many or all FOXF1 target genes as a set is also informatically possible, and may reveal a striking difference. I think that evaluation of DNA repair genes (either a handful of candidates or as a set) could also reveal interesting results, as I suspect that DNA damage as shown in the comet assays (bravo) would induce stress or repair systems. The paper implies that such stress or repair systems are insufficient in PAH in part due to FOXF1. Such phenomena could be dissected from the gene expression data onhand with more precision than KEGG or GO would yield (without additional experiments).

POINT-BY POINT RESPONSE TO THE REVIEWER COMMENTS

Reviewer #1 (Remarks to the Author):

Dear colleagues,

I have read with interest your paper on role of ATM BMRP2 and Foxf1 in endothelial cells DNA damage signalling in pulmonary hypertension.

I found the paper very well written logical and easy to follow. I have several minors comments.

We are pleased that the reviewer found our paper “interesting” and “very well-written” and we greatly appreciate the constructive suggestions that have resulted in new experiments and clarifications that have strengthened our findings and placed them in important context.

1) given the model used hypoxia followed by reoxygenation I suppose that authors believe that DNA damage results from oxidative damage due to ROS? Could authors monitor ROS levels during the course of the experiments?

Response 1: The reviewer makes an important point. To address it, we reduced *ATM* or *BMPR2* in human PAEC by siRNA, and used non-targeting siRNA as a control, and measured ROS levels using DCFDA during normoxia, hypoxia, and reoxygenation at 15, 30 and 60 minutes. Interestingly, while there was a small increase in ROS during hypoxia, it was reoxygenation as short as 15 minutes that produced a massive increase in ROS, which then decreased over time. While loss of ATM and BMPR2 contributed to the increase in ROS, the effect was minor relative to the absolute increase in ROS, particularly with reoxygenation. This suggests that loss of ATM or BMPR2 impact the repair of DNA damage by a mechanism largely independent of the relatively minor differences in ROS produced by the different genotypes. The data are presented in **Supplementary Figure 2d** and discussed in the Results, page 8, line 158-167.

2) given that multiple DNA damage pathways are affected in PHT it will be interesting to assess the expression levels of couple of them including DNA PK which as just been showed to be important in PAH (January 2023 paper in the red ATS journal)

Response 2: To address this question, we investigated the other DNA damage response enzymes, ATR and DNA-PK, in human PAEC transfected with si*ATM* and si*BMPR2*. Reduction of *ATM* and *BMPR2* upregulated ATR and DNA-PK in normoxia and reoxygenation. Despite the upregulation of these pathways, DNA damage, judged by an increase in γ H2AX or pRPA2 is more persistent with loss of ATM or BMPR2. The data are presented in **Supplementary Figure 2c**, and discussed in the Results, page 7 line 152- page 8 line 157, and in the Discussion, page 16 line 360- page 17 line 362.

3) can authors speculate what drives FOXF1 down regulation? In PAH tissues is it specific to EC?

Response 3: We have previously shown decreased p53 in PAEC in EC-*Bmpr2*^{-/-} mice after reoxygenation. That result was consistent with the data in Figure 4f. *Foxf1* has been reported to be regulated by p53 (ref. 30), and decreased P53 after reoxygenation with loss of *Bmpr2* or *Atm* presumably decreases *Foxf1*. In lung sections from patients with PAH, FOXF1 is expressed in the nuclei of vWF-positive vascular endothelial cells, but not in SMC or non-EC or non-SMC cells in the vascular lesion, as we show in **Figure 6b** and **Supplementary Figure 6d** and discuss in the Discussion, page 18 line 401-406.

4) given that Foxf1 regulates fanconi complex are the members of the complex affected in EC

Response 4: This is a very important point. Our bulk RNA seq of mouse lung EC showed that *Fancc* is decreased in both EC-*Atm*^{-/-} reoxy and EC-*Bmpr2*^{-/-} reoxy compared to control reoxy. FOXF1 interacts with multiple Fanconi anemia proteins, and loss of FOXF1 in HUVEC leads to reduced FA protein stabilization (ref. 28). FANCC forms a complex with multiple FA proteins such as FANCM to activate BRCA1 and RAD51 to repair DNA damage (ref. 60). Given that BRCA1 and RAD51 are

targets of BMPR2 signaling (ref. 14, 15), decreased activity of RAD51 and BRCA1 is likely induced by decreased FA complex formation following loss of FANCC with reduced BMPR2 or ATM. The data are presented in **Figure 4 c and d**, and in the Results, page 11, line 227-234 and in the Discussion, page 19 line 408-415.

5) why using hypoxic mice followed with reoxygenation and not sugen ? This experimental model seems to be closer to group 3 PHT than group 1 according to Boucherat review in Circulation research on animal model.

Response 5: This is an important distinction. Group 3 PAH is caused by lung disease and hypoxia, but our model is not hypoxia per se or lung disease. Here we use hypoxia as a second hit in two genetic models. Moreover, our models largely reflect extensive DNA damage that occurs due to a major increase in oxidative stress induced by reoxygenation. Deficiency of *Bmpr2* or *Atm* impairs DNA damage repair in both genotypes, indicating that accumulation of unrepaired DNA damage is linked to persistent PAH by a common pathway.

We and other investigators have found that the Sugen/Hypoxia mouse model does not cause robust pulmonary hypertension, and is reversible when mice are returned to normoxia as opposed to the response of the rat to Sugen/Hypoxia (Toba et al. doi: [10.1152/ajpheart.00728.2013](https://doi.org/10.1152/ajpheart.00728.2013)). However, it is interesting that in our models, expression of *Vegfr2* was decreased with severe oxidative stress when reoxygenation (not hypoxia) was added to loss of *Bmpr2*, the most common genetic mutation in Group 1 PAH. This is now addressed in the Discussion, page 17 line 364-369.

6) is there any alteration in DNA damage response in ECs isolated from systemic vessels or is it specific to the lungs because of hypoxia and reoxygenation? Any systemic abnormalities

Response 6: This is an important point. We therefore examined DNA damage in the endothelium of the aorta, renal arteries, and coronary arteries in EC-*Bmpr2*^{-/-} and control mice after reoxygenation. In the systemic vascular EC of EC-*Bmpr2*^{-/-} mice, γ H2AX foci were similar to those in Control mice. In addition, we performed blood tests to check for evidence of functional impairment in other organs. Although we found a few statistically significant differences in EC-*Bmpr2*^{-/-} mice compared to controls, all values were within normal limits for mice. The data are presented in **Supplementary Figure 1i and j**, and discussed in the Results, page 7 line 133-139.

7) although mice models are EC specific any alteration in PASMC or fibroblasts? Just wondering here if the presence of damage ECs can affect the functioning of the surrounding PASMCs.

Response 7: The reviewer's point is well taken. Although we deleted *Bmpr2* in EC there could be paracrine effects in other vascular cells. Indeed, we found increased DNA damage in SMC of EC-*Bmpr2*^{-/-} mice in normoxia and a further increase after reoxygenation. We previously showed that loss of *Bmpr2* in EC induces an increase in inflammatory cytokines that can have deleterious effects on neighboring SMC (ref. 7). The data are presented in **Figure 1f** and discussed in the Discussion, page 17 line 380- 383.

8) as much as possible try to reach sample size of greater than 5,

Response 8: At the reviewers' suggestion, we increased the n for Immunostaining to 6 (**Figures 1e, 2e, 6a, 7e, 7f;** and **Supplementary figures 1h, 2h**).

9) comet assay should be presented by the mean of tail length by patients or mice not the total cell count which will artificially increase power.

Response 9: We now present data as mean values for the comet assay in PAH patients in **Supplementary Figure 6j**. We did not perform the comet assay in murine cells.

Reviewer #2 (Remarks to the Author):

In this manuscript, Isobe et al investigated a causal relationship between DNA damage and the development of pulmonary arterial hypertension (PAH). The premise stems from a large set of previously published work that links BMPR2 to DNA damage repair. In the current study, the authors found that FOXF1 transcription factor is decreased in endothelial cells from 2 different animal models that use the exposure to hypoxia-reoxygenation cycle to cause PAH. Importantly, the authors demonstrated that FOXF1 gene replacement therapy was sufficient to ameliorate the pulmonary hypertensive phenotype in both animal models. I commend the authors for the amount of work performed in this project and congratulate them for generating such a comprehensive study with significant clinical implications. The manuscript contains high-quality experimental data, and main conclusions are supported by the data. However, the manuscript can be improved by addressing the following issues:

We greatly appreciate the remarks of Reviewer #2, the acknowledgement of 'the amount of work in generating our comprehensive study with significant clinical implications', the 'high quality experimental data' and 'main conclusions supported by the data'. The suggestions made have resulted in new experimental data that have extended and clarified and strengthened our findings.

Major points:

1. Do ATM^{-/-} mice exhibit similar vascular remodeling compared to BMPR2 mice? Do they develop vessel obliteration or any other type of complex vascular lesions?

Response1. As with EC-*Bmpr2*^{-/-} mice, EC-*Atm*^{-/-} mice showed increased muscularization of pre-capillary arteries as well as a decreased number of those distal vessels at alveolar duct and wall level, but no other vascular abnormalities such as plexiform lesions, occluded vessels or AVMs. Representative histology sections of EC-*Atm*^{-/-} mice are now shown in **Figure 3c** and discussed in Results, page 9 line 192-194.

2. Is there reduced FOXF1 expression at baseline in both models? If not at baseline, can the authors speculate or provide the data to demonstrate at which time point FOXF1 downregulation occurs?

Response 2: We appreciate being directed to address this point. While previously we showed reduced FOXF1 intensity in the mice with EC-*Bmpr2*^{-/-} mice in normoxia. In **Supplementary Figure 6a**, we now add data showing reduced FOXF1 in EC-*Atm*^{-/-} mice during normoxia. We also include new data in **Supplementary Figure 6c** showing reduced mRNA expression of *Foxf1* in cultured PAEC in the EC-*Atm*^{-/-} and EC-*Bmpr2*^{-/-} mice in normoxia. The increase in DNA damage was therefore likely related to reduced *Foxf1* expression in normoxia as well as reoxygenation. The data are presented and discussed in the Results, page 13 line 273-275 and page 13 line 276-278.

3. Please provide a scatter plot correlating RVSP with RV/LV+S. Showing group data is useful for intergroup comparisons however it does not allow to see the linear relationship that is expected from the experiment.

Response 3: We added scatter plots correlating RVSP and RV/LV+S. Since $R^2 > 0.6$ and $p < 0.05$, it is considered that RVSP and RV/LV+S have a linear relationship. The data are presented in **Supplemental Figures 3a, 3b, 3e, 7d**.

4. Is there a difference in FOXF1 expression between TdT positive versus negative cells? Having single cell sequencing data provides an opportunity to truly evaluate the transcriptional profile differences between lacking BMPR2 (positive Cre recombination, TdT positive) and cells with presumably normal BMPR2 (TdT negative).

Response 4: To answer this question, we determined whether tdTomato negative EC exist in EC-Bmpr2^{-/-} mice following reoxygenation. We found 219 cells that expressed endothelial markers *Cd31* (*Pecam1*) and *Cldn5* (*Pecam1* > 0 and *Cldn5* > 0) that were tdTomato negative (value=0). The expression in these cells was compared with tdTomato positive (*Bmpr2* deleted) cells in the EC-Bmpr2^{-/-} reoxy mice (n=12,469). *Cldn5* was significantly decreased in the tdTomato-negative cells, suggesting that in addition to the paucity of number they are also not comparable to the tdTomato-positive cells to address the question. Moreover, levels of *Bmpr2* are also reduced and *Foxf1* is even lower than in the tdTomato-positive ECs. These cells are therefore not *Bmpr2*-positive ECs for comparison and could represent a rare transitional cell. This is discussed in the Methods, page 28 line 610-612.

sc-RNA seq for lung CD31+ CD45- cell in EC-bmpr2^{-/-} reoxy mice

5. Flow cytometry data are needed to show specific targeting of endothelial cells and off-target cell selection. AAV with GFP in wild type s/p hypoxia mice should be sufficient. IF alone is insufficient evidence.

Response 5: To determine whether there was targeting of AAV to non-endothelial cells, we applied flow cytometry to enzymatically digested whole mouse lungs following transfection with AAV carrying luciferase, Luciferase-positive cells delivered by this AAV vector included almost no tdTomato-negative cells (0.83% of the total lung digest), compared to 24.7% of the total lung digest that was luciferase positive and tdTomato-positive (= EC). The data are presented in **Supplementary figure 7a, b** and discussed in the Results, page 14 line 315- page 15 line 319.

6. It is possible that the phenotype after AAV-Foxf1 treatment is secondary to new vessel formation (thus dropping pulmonary vascular resistance) rather than reversal of established pathological vascular remodeling. This possibility should be addressed in the Discussion section.

Response 6: We found that delivery of FOXF1 improved PAH and improved the number of blood vessels, and we agree that this may be due to the regeneration of new blood vessels rather than the repair of originally damaged blood vessels. thus at least in part the cause of the reduced percentage of muscularized distal arteries. We added this in the Results, page 15 line 324-326.

7. It will be helpful to provide additional data on RV function (either TAPSE or CO) for treatment groups.

Response 7: We added the cardiac output data in **Supplementary Figures 3c, 3d and 7c**.

Minor points:

1. Page 6 line 124: It is unclear if *Cdh5-CreER/Rosa-TdTom* mice were treated with tamoxifen and used as controls or the mice without tamoxifen treatment were used as control. Please specify.

Response 1: We used *Cdh5-CreER/Rosa-TdTomato* mice treated with tamoxifen as controls. This is described in the Methods (page 22 line 483-484), and we now added it to the Results, page 6 line 126 and page 8 line 173.

2. There is a mistake (Line 80, page 4, introduction): "...persistent DNA damage in ECs and pulmonary arterial hypertension resultS from.."

Response 2: Thank you for the comment. We clarified the sentence – it now reads:

"However, it has been unclear whether persistent DNA damage in ECs and pulmonary arterial hypertension both result from a common mechanism, which could inform new therapeutic avenues." (page 4 line 82)

3. Page 6, line 121: The reference for the mouse model cited goes to a paper where the mouse model is cited to a different paper by Spiekerkoetter. The mouse model by Spiekerkoetter in JCI paper is a *SCL-CreERTM+/R26R/Bmpr2-/-*. The authors state they are using a *Cdh5-CreER/Rosa-TdTom*. These are effectively 2 different models. Please, correct this error?

Response 3: Thank you for pointing this out. You are correct, we used *Cdh5-Cre* mice as the endothelial cell driver in the studies presented in this paper. To clarify the difference from the mouse used by Spiekerkoetter et al. (ref. 21), the text has been corrected (Please see page 4 line 78).

4. I commend the authors for generating a visually appealing graphical abstract. However, some information is misleading. As portrayed, it appears as if PAH secondary to *BMPR2* dysfunction results from capillary abnormalities and perhaps arterialization of veins or even A-V shunts which perhaps is more common in ACDMPV rather than PAH.

Response 4: Thank you for your comment. We modified the graphical abstract more clearly represent our findings. Please see the abstract figure.

Reviewer #3 (Remarks to the Author):

General comments:

In this study, Isobe et al. investigated the cellular mechanisms underlying persistent pulmonary hypertension (chronic

hypoxia-induced PH that persisting after reoxygenation) in endothelial cells (EC) specific-Bmpr2^{-/-} mice. In those mice, reoxygenation after chronic hypoxia, in contrast to control mice, did not reverse pulmonary hypertension, which was associated with increased DNA damage. The deletion of *Atm* in EC (EC-*Atm*^{-/-} mice) mimicked the EC-Bmpr2^{-/-} mice phenotype after reoxygenation. Transcriptome analysis showed that downregulation of the transcriptional factor *Foxf1* can be cause of persistent pulmonary hypertension in EC-Bmpr2^{-/-} and EC-*Atm*^{-/-} mice. Restoration of *Foxf1* level during reoxygenation reversed persistent pulmonary hypertension. These findings are very interesting.

We appreciate the reviewer's comments in indicating that our 'findings are very interesting', and greatly appreciate the comments and suggestions that have extended, reinforced and clarified our findings.

Major comments:

1. Is there a difference between *Foxf1* and/or *Atm* expression between EC of PH patients with and without *BMPR2* mutation. Your data claim that *Atm* and *Foxf1* is linked to *BMPR2* – Thus, a difference in their expression between +/-*BMPR2* mutations in PH patients would be expected.

Response 1: To address this question, we performed immunostaining of lung sections of IPAH patients with or without a *BMPR2* mutation. *FOXF1* intensity in the EC of the pulmonary vascular lesions in lungs of patients without a *BMPR2* mutation was decreased to the same degree as in lungs of PAH patients carrying a *BMPR2* mutation. The data are shown in Figure 6b. This could be explained by previous reports indicating that *BMPR2* is reduced in IPAH patients that do not carry a mutation (ref. 4), or it could indicate that reduced *FOXF1* may be a common feature of IPAH as well as HPAH. We also showed that cultured PAEC from PAH patients with or without a *BMPR2* mutation have decreased *FOXF1* expression (Supplementary Figure 6e). This is discussed in Results page 13 line 281- 285.

2. Have statistical comparisons between all groups in the in vivo experiments performed? E. g. Figure a, b: it is not clear if a difference exists between "normoxia", "hypoxia" and "reoxygenation" within one genotype. As an example, the muscularization in figure 3a for the *Atm* knockout mice seems to be decreased in reoxygenation compared to hypoxia. This looks similar for all other parameters of PH. In this regard it would be interesting to know if PH indeed persists in reoxygenation or if the recovery is delayed only. What would happen, if you follow the mice upon reoxygenation for a longer duration? This is critical to judge on the interpretation if PH is indeed persistent or recovery is delayed only.

Response 2: Thank you for your important question regarding persistence of pulmonary hypertension following reoxygenation vs. delayed recovery. Statistical analysis between all groups were performed by two-way ANOVA. At one month there is some reduction in RVSP and muscularization over the hypoxia value in the EC-*Atm*^{-/-} mice, begging the question that there could be a delay in resolution. To investigate whether PH persists in EC-Bmpr2^{-/-} and EC-*Atm*^{-/-} mice after a longer period of reoxygenation or 'recovery' in room air, we extended the experiment and evaluated pulmonary hypertension, judged by RVSP and RVH after two months of reoxygenation. We saw no further regression of pulmonary hypertension in EC-Bmpr2^{-/-} and EC-*Atm*^{-/-} mice after two months of reoxygenation. The data are presented in Supplementary Figure 3e, f, and g, and accompanying Results, page 9 line 192- 199.

3. Why are there largely differing number between groups. As an example, in figure 3 you show n=4-8 – the hypoxia *Atm* group consist of only n=4, the respective WT has a higher number and this differs from the reoxygenation groups. Please state for all in vivo mouse experiments how many animals were allocated to which group at the beginning of the experiment and if the group size is different, why.

Response 3: The reviewer is correct. In hypoxia, n=6 male control and n=4 male EC-*Atm*^{-/-} whereas the female cohort has an n=6 control and n=5 EC-*Atm*^{-/-}. We did not pursue larger cohorts because the emphasis was on the phenotype during reoxygenation, and we observed no differences in the DNA damage response during hypoxia. There was no attrition in our experiments and all mice were included in the data analysis. This is clarified in the Methods, page 21 lines 471- page 22 line 475.

4. The authors demonstrated that hypoxia alone (without reoxygenation) induced DNA damage of endothelial cells (proven by a higher level of H2AX and comet tail length in Supplementary figure 1 and 2) - it will be interesting to investigate the level and role of Foxf1 in EC during chronic hypoxia. Does the downregulation of Foxf1 play a role not only in the repair mechanism during reoxygenation after exposure to chronic hypoxia, or also in development of chronic hypoxia-induced PH?

Response 4: The reviewer raises a very interesting point. To answer it, we performed immunostaining on control and EC-*Bmpr2*^{-/-} mice under normoxia and chronic hypoxia and found a similar reduction in the intensity of FOXF1 in both control and EC-*Bmpr2*^{-/-} mice during hypoxia compared to control normoxia. Consistent with this, *Foxf1* mRNA expression was similarly decreased in cultured PAEC from EC-*Bmpr2*^{-/-} and control hypoxic mice compared to control normoxic mice. These data are provided in **Supplementary Figure 6 b, c** and discussed on page 13 line 274-279. As the reviewer points out, DNA damage during hypoxia may be the result of low FOXF1.

5. Please show the basal level of Foxf1 and Atm in EC isolated from patients with PAH compared with control EC.

Response 5: We examined *FOXF1* and *ATM* gene expression in lung EC isolated from 6 PAH patients (3 patients carrying a *BMPR2* mutation, and 3 patients without a *BMPR2* mutation) and 6 controls. Expression of both *FOXF1* and *ATM* were reduced in all PAH patients. These data are provided in **Supplementary Figure 6e** and discussed on page 13 line 281-283.

6. The spatial expression of Atm in PAH is not clear. Does Atm expression change in PAH? In which cells did Atm expression change in PAH?

Response 6: Unfortunately, we could not find antibodies that allowed us to detect ATM immunostaining of lung sections in murine or human tissues, and our single cell RNA-seq was only performed in lung ECs. We could not exclude a reduced expression in other cells of the vessel wall, although the deletion of *BMPR2* or *ATM* was EC-specific.

7. Why was 0.5 %O₂ used for EC culture. This does not reflect the in vivo situation of EC pO₂. Is there a difference if you use physiological relevant pO₂ levels.

Response 7: The reviewer is correct in that pO₂ levels are more important physiologically than the percentage of oxygen administered, but the cell culture system is actually a model of relative hypoxia. The value generally used as the physiological pulmonary arterial pO₂ level (mixed venous oxygen tension; PvO₂) is 40 mmHg (ref. 69). To be sure that we are in physiologic pO₂ range, we measured pO₂ in the EC culture medium inside the 0.5% O₂ hypoxia chamber. The mean pO₂ level of the culture medium in the hypoxia chamber is 36.0 mmHg at 24 hours and 37.6 mmHg at 48 hours, whereas in normoxia it is 150.4 mmHg at 24 hours and 145.7 mmHg at 48 hours (measured by Siemens, RAPIDPoint 500 Systems). Normal oxygenated blood PaO₂ at 100% saturation is 128 mmHg. This indicates that even with 0.5% FiO₂ we are measuring a relative hypoxia in our cells. We have clarified this in the Methods, page 20 line 452- page 21 line 456.

While the pO₂ level of our culture medium was modestly lower than physiological mixed venous oxygen, we show that reoxygenation, not hypoxia per se, triggers a major increase in ROS (**Supplemental Figure 2d**), leading to impaired DNA damage repair in the cells with reduced ATM or *BMPR2*. Furthermore, to investigate whether the cell culture results are meaningful in terms of PAH pathology, we used an animal model and showed that persistent DNA damage caused by reoxygenation in association with a deletion of ATM or *BMPR2* in EC is related to reduced FOXF1, as an explanation for persistent pulmonary hypertension.

Interestingly an SvO₂ <65%, is used as a prognostic indicator in PAH (Khirfan et al. doi: 10.1016/j.chest.2020.06.053). An SvO₂ of 65% corresponds to a PvO₂ of 35 mmHg according to the oxygen dissociation curve in the normal state (Kawakami et al. doi: 10.1056/NEJM198305053081801). PvO₂ <35mmHg is associated with a poor prognosis of PAH (Nagata et al. doi: 10.1186/s12890-022-02073-0).

Minor comments

1. To analyze the data from the WB (e.g. Figure 1a) it seems normalization was performed. The normalization protocol must be described in methods or in labelling of x-axis.

Response 1: In this experiment we investigated how the effect of loss of *Bmpr2* or reoxygenation compared to the normoxia control. We repeated the experiment three times, so we quantified the value relative to the normoxic control for each experiment. We now include this explanation under "Methods", page 33 line 722-726.

2. If you give the hypoxia level in %, please add the information (e.g. in the methods) at which altitude the experiments were performed.

Response 2: All animal and cell culture experiments were performed at sea level. 10% oxygen represents the hypoxic oxygen tension at an altitude of 19,000 ft (5791m). This is indicated in the Methods, page 20 line 451-452 and page 22 line 487- 489.

Reviewer #4 (Remarks to the Author):

The manuscript by Isobe and colleagues is a very nice paper that seeks to establish a role for the FOXF1 transcription factor in the correction of Pulmonary Artery Hypertension (PAH) with a further implication that DNA damage is among the molecular pathological insults that drive disease. To this end, they have used 2 separate animal models of PAH as well as multiple cellular studies with Pulmonary Artery Endothelial cells from diseased and control sources. They describe the consistency of FOXF1 downregulation with PAH induction as well as the extent to which restoration of FOXF1 activity via transfection can rescue the dysfunctional cellular physiology in EC cells as well as reversal of the PAH pathophysiology in their animal models.

Major comments:

None

Minor comments:

There is an opportunity to further explore the gene expression data with a more directed focus on FOXF1, which may add weight to the proposition of FOXF1's role. Both Seq and single cell datasets are often rich beyond our ability to make meaningful interpretations, and the authors provide substantiation for the role of FOXF1 from a handful of candidate target genes. However, the interrogation of many or all FOXF1 target genes as a set is also informatically possible, and may reveal a striking difference. I think that evaluation of DNA repair genes (either a handful of candidates or as a set) could also reveal interesting results, as I suspect that DNA damage as shown in the comet assays (bravo) would induce stress or repair systems. The paper implies that such stress or repair systems are insufficient in PAH in part due to FOXF1. Such phenomena could be dissected from the gene expression data on hand with more precision than KEGG or GO would yield (without additional experiments).

Response: The reviewer's point is certainly well taken. We therefore searched our bulk RNA seq data for any differentially expressed genes that were *Foxf1* target genes and related to DNA repair. Using publicly available lung ChIP-seq data, we found 11,222 genes with *Foxf1* peaks at a distance of +/-1 kb or less from the TSS. We also used a gene list associated with DNA repair in mice (GO 0006281) showing 577 genes. Comparing these gene lists, 493 genes are both *Foxf1* targets and associated with DNA repair. In our bulk RNA seq data, 11 genes, including *Fancc*, *Rad51ap1* and *Rad21*, were downregulated in EC-*Bmpr2*^{-/-} mice following reoxygenation. It has been reported that FACC forms a complex with other FA proteins, and this complex activates *Brca1* and *Rad51* and repairs DNA damage (ref. 60) and *Brca1* and *Rad51* are known downstream targets of *Bmpr2* signaling (ref. 14,15). These results are now included in the description of the RNA seq experiment, see Results page 11 line 227- 234 and Discussion page19 line 408-415.

REVIEWERS' COMMENTS

Reviewer #1 (Remarks to the Author):

No further comments excellent work

Reviewer #2 (Remarks to the Author):

My comments were addressed.

Reviewer #3 (Remarks to the Author):

The authors have comprehensively and conclusively addressed all the questions I raised. I have only two minor comments:

1. In the Supplementary figure 2d you state: "However, the major increase in ROS levels in all three cell types occurred during the first 15 minutes of reoxygenation and was mostly resolved during the first hour (Supplementary Fig. 2d), with ROS production only slightly increased with reduced ATM or BMRP2 at each time point.

However, in the right upper panel no statistics is given for this statement (increase during reoxygenation). Please provide statistics.

2. The Figure 3 legend text is misleading:

EC-Atm^{-/-} and control mice were subjected to 3 weeks of hypoxia (10% O₂) followed by 4 weeks of reoxygenation in room air (reoxy), 7 weeks of room air (normo), or 3 weeks of hypoxia (10% O₂, Hx).

This can be interpreted as if the mice were exposed for 3 weeks to hypoxia and then they either were kept a) for 4 weeks in normoxia or b) were kept for 7 weeks in normoxia after 3 weeks of hypoxia or c) were kept for another 3 weeks in hypoxia after 3 weeks of hypoxia.

I suggest changing this to: EC-Atm^{-/-} and control mice were subjected to a) 7 weeks of room air (Normo), or b) 3 weeks of hypoxia (10% O₂, Hx) only, or c) 3 weeks of hypoxia (10%O₂), followed by 4 weeks of reoxygenation in normoxia (Reoxy)

Reviewer #4 (Remarks to the Author):

The authors have addressed my comments sufficiently.

POINT-BY POINT RESPONSE TO THE REVIEWERS' COMMENTS:

Reviewer #1

No further comments excellent work

We thank Reviewer 1 for the kind comments and appreciate that the findings of our manuscript have been strengthened by the additional experiments suggested and included in the revision.

Reviewer #2

My comments were addressed.

We are delighted that the revised document addressed all points raised by Reviewer #2, and also feel that the revisions strengthened our findings and improved the manuscript.

Reviewer #3 (Remarks to the Author):

The authors have comprehensively and conclusively addressed all the questions I raised. I have only two minor comments:

1. In the Supplementary figure 2d you state: "However, the major increase in ROS levels in all three cell types occurred during the first 15 minutes of reoxygenation and was mostly resolved during the first hour (Supplementary Fig. 2d), with ROS production only slightly increased with reduced ATM or BMRP2 at each time point. However, in the right upper panel no statistics is given for this statement (increase during reoxygenation). Please provide statistics.

We apologize for the confusion. The timeline on the top right is a different representation of the data on the bottom bar graph, and the latter was used for statistical analysis. We clarified this in the legend to Supplementary Fig. 2d.

2. The Figure 3 legend text is misleading:

EC-Atm^{-/-} and control mice were subjected to 3 weeks of hypoxia (10% O₂) followed by 4 weeks of reoxygenation in room air (reoxy), 7 weeks of room air (normo), or 3 weeks of hypoxia (10% O₂, Hx).

This can be interpreted as if the mice were exposed for 3 weeks to hypoxia and then they either were kept a) for 4 weeks in normoxia or b) were kept for 7 weeks in normoxia after 3 weeks of hypoxia or c) were kept for another 3 weeks in hypoxia after 3 weeks of hypoxia.

I suggest changing this to: EC-Atm^{-/-} and control mice were subjected to a) 7 weeks of room air (Normo), or b) 3 weeks of hypoxia (10% O₂, Hx) only, or c) 3 weeks of hypoxia (10%O₂), followed by 4 weeks of reoxygenation in normoxia (Reoxy)

We appreciate the reviewer's suggestion, that has now been incorporated.

Reviewer #4

The authors have addressed my comments sufficiently.

We are delighted that the revised document addressed all points raised. We appreciate that these revisions strengthened our findings and improved our manuscript.